# Exercise enhances skeletal muscle regeneration by promoting senescence in fibro-adipogenic progenitors

Yuki Saito ⬤ [1], Takako S. Chikenji ⬤ [1,2✉], Takashi Matsumura[3], Masako Nakano[1] & Mineko Fujimiya[1]

Idiopathic inflammatory myopathies cause progressive muscle weakness and degeneration. Since high-dose glucocorticoids might not lead to full recovery of muscle function, physical exercise is also an important intervention, but some exercises exacerbate chronic inflammation and muscle fibrosis. It is unknown how physical exercise can have both beneficial and detrimental effects in chronic myopathy. Here we show that senescence of fibro-adipogenic progenitors (FAPs) in response to exercise-induced muscle damage is needed to establish a state of regenerative inflammation that induces muscle regeneration. In chronic inflammatory myopathy model mice, exercise does not promote FAP senescence or resistance against tumor necrosis factor–mediated apoptosis. Pro-senescent intervention combining exercise and pharmacological AMPK activation reverses FAP apoptosis resistance and improves muscle function and regeneration. Our results demonstrate that the absence of FAP senescence after exercise leads to muscle degeneration with FAP accumulation. FAP-targeted pro-senescent interventions with exercise and pharmacological AMPK activation may constitute a therapeutic strategy for chronic inflammatory myopathy.

[1] Department of Anatomy, Sapporo Medical University School of Medicine, Sapporo, Japan. [2] Department of Health Sciences, School of Medicine, Hokkaido University, Sapporo, Japan. [3] Department of Orthopaedic surgery, Sapporo Medical University School of Medicine, Sapporo, Japan. ✉email: chikenji@pop. med.hokudai.ac.jp

diopathic inflammatory myopathies are rare, heterogeneous, autoimmune diseases featuring muscle weakness, inflammation, and fibrosis[1]. The pathogenetic mechanism of these myopathies is unknown, but autoimmune processes are strongly implicated[1]. Treatments for inflammatory myopathies include oral prednisolone and methotrexate. Recently, it has been recognized that physical exercise is also a potentially powerful therapy for these conditions, but it is not likely to provide beneficial effects to all patients, and it is therefore difficult to perform aggressive interventions for idiopathic myopathy[2]. There are concerns that excessive physical exercise could result in significant muscle damage, thus causing acute renal failure, liver dysfunction, compartment syndrome, heart failure, arrhythmias, electrolyte imbalance, and in severe cases, death[3]. It has also been shown that excessive exercise can induce fibrosis in cases of chronic inflammatory myopathy (CIM)[4,5]. It is still largely unknown how physical exercise can play both positive and negative roles in chronic myopathy.

Fibro-adipogenic progenitors (FAPs), also known as muscle-resident, platelet-derived growth factor receptor-α-positive (PDGFRα+) mesenchymal progenitors that are distinct from muscle stem cells, are key regulators of muscle regeneration and homeostasis, but also cause chronic inflammation and fibrosis when aberrantly regulated[6–8]. Damage to skeletal muscle, whether due to myotoxic, chemical, or mechanical sources, triggers a transient phase of FAP proliferation and protein expression that promotes muscle stem cell differentiation, followed by a return to baseline with apoptosis and subsequent phagocytic clearance[9,10]. In chronic muscle inflammation, FAP proliferation and protein expression are maintained at high levels, and it is difficult to induce apoptosis and remove dead cells via phagocytosis; this results in the accumulation of FAPs, which leads to muscle fibrosis and continued low-grade inflammation[10]. In normal and α7-integrin transgenic mice, physical exercise also triggers FAP proliferation and muscle regeneration that are related to protein expression[11], but the effects of exercise on FAPs in chronic muscle inflammation are still unknown. As described above, FAPs have two phenotypes: one is pro-regenerative and pro-apoptotic, while the other is pro-fibrotic and anti-apoptotic. We hypothesize that the effects of exercise on muscle regeneration or fibrosis depend on the FAP phenotype.

Here we show that FAPs in regenerating muscle, but not degenerating muscle, acquire features of senescence, which is an important phenomenon related to both cell apoptosis and clearance[11]. FAPs in exercised normal muscle also acquire senescent features, but this change is not observed in exercised mice in a chronic myopathy model. The compound 5-aminoimidazole-4-carboxamide-1-β-D-ribofuranoside (AICAR), a cell-permeable APMK (AMP-activated protein kinase) activator with previously reported pro-senescent effects[12], restores FAP senescence and promotes apoptosis in chronic myopathy, and dramatically increases the therapeutic effects of exercise in cases of chronic myopathy. Our results reveal a mechanism by which FAP acquisition of senescent features has positive effects on muscle regeneration and prevents muscle fibrosis.

## Results

### Accumulation of FAPs with apoptosis resistance in CIM.
We used two mouse models to investigate the differences in FAP characteristics during regenerative and degenerative processes in the muscle. To replicate regenerating muscle, we created an acute muscle injury (AMI) model by intramuscular injection 50 μl of barium chloride (1.2% in sterile distilled water), and CIM mice were used as the degenerative muscle model (Fig. 1a, b and Supplementary Fig. 1A–D)[13]. We first identified an increase in the number of FAPs defined as PDGFRα+ interstitial cells in both AMI and CIM muscle compared with control muscle (Fig. 1c, d), and the fibrotic area with collagen deposition was larger in CIM muscle than in both control and AMI muscle (Fig. 1c, e). The number of FAPs that were lineage-negative (Lin−), CD31−, α7-integrin−, and PDGFRα+ (Fig. 1f) increased but returned to pre-damage levels after 7 days in AMI mice; however, in CIM mice, these cells continued to accumulate for 14 days after the onset (Fig. 1g). As this decrease in FAPs may be a crucial step in the initiation of regeneration, we investigated whether FAPs undergo apoptosis following tumor necrosis factor-α (TNF-α) stimulation, which is a key factor in FAP death[10]. In AMI mice, lactate dehydrogenase (LDH) release from FAPs and the percentage of annexin V-expressing apoptotic FAPs were both increased by TNF-α stimulation; however, these increases were not observed in CIM-FAPs (Fig. 2a–c). Furthermore, PrimerArray® analysis showed that the gene expression of TNF receptor superfamilies, such as *Fas*, *Cd27*, *Tnfrsf14*, and *Tnfrsf19*[14], was increased in AMI-FAPs and decreased in CIM-FAPs, while the expression of tumor growth factor-β (TGF-β)/BMP superfamily members, *Amhr2* and *Csf1*, was increased in CIM-FAPs (Fig. 2d, e). AMH signaling regulates BMP, Akt, NF-κB (nuclear factor-κB), and SMAD signaling in non-small-cell lung cancer cells, and influences cell survival[15]. Csf1 promotes M2-like skewing of macrophage function and inhibits TNF-α expression[16], and its expression by fibroblasts maintains the stability of macrophage and fibroblast cell populations[17]. CC-chemokine ligands, which are important in recruiting macrophages and promoting muscle regeneration[18], were also increased in AMI-FAPs (Fig. 2d, e). These results indicated that muscle damage induced a pro-inflammatory and pro-apoptotic phenotype that enhanced muscle regeneration in AMI-FAPs, but CIM-FAPs demonstrated an anti-apoptotic phenotype that was not pro-inflammatory. In fact, the percentage of active caspase-3-positive FAPs were increased in the muscles of AMI mice, but not in those of CIM mice (Fig. 2f, g), and the percentage of CD11b+ cells, including monocytes, macrophages, NK cells, and granulocytes, was higher in AMI mice than in CIM mice (Supplementary Fig. 2A, B). We also found that CIM-FAPs expressed Bcl-2, an anti-apoptotic marker, and exhibited decreased expression of p53, a key molecule in regulating apoptosis and cell senescence[11] (Fig. 2h). Further, we investigated the messenger RNA (mRNA) expression levels of *Cd274*, *Pdcd1lg2*, and *Cd47*, which encode programmed death ligand 1 and 2 (PD-L1 and PD-L2) and CD47. Each of these molecules is responsible for crucial immune system signaling: in the cases of PD-L1 and PD-L2, the "don't find me" signal, and in the case of CD47, the "don't eat me" signal. The upregulation of these signals enables cells to adapt to the immune system and to avoid phagocytosis[19–21]. The mRNA expression levels of *Cd274*, *Pdcd1lg2*, and *Cd47* in CIM-FAPs were significantly higher than in AMI-FAPs (Fig. 2i), and we found that these levels were strongly correlated with the mRNA expression of *Cdkn2a*, a senescence marker in FAPs[11] (Fig. 2j).

### FAPs acquire senescent features after acute muscle injury.
To understand the mechanism whereby CIM-FAPs acquire apoptosis resistance and clearance deficiency, we next evaluated the degree of senescence seen in FAPs, because senescence is an important phenomenon related to both cell apoptosis and clearance[11]. We compared the mRNA expression of senescence factors *Cdkn2a*, *Trp53*, *P21*, and *p19Arf* in freshly isolated FAPs from AMI, CIM, and control mice. The mRNA expression levels of *Cdkn2a* and *Trp53* were significantly decreased in CIM-FAPs compared with control-FAPs and AMI-FAPs (Fig. 3a).

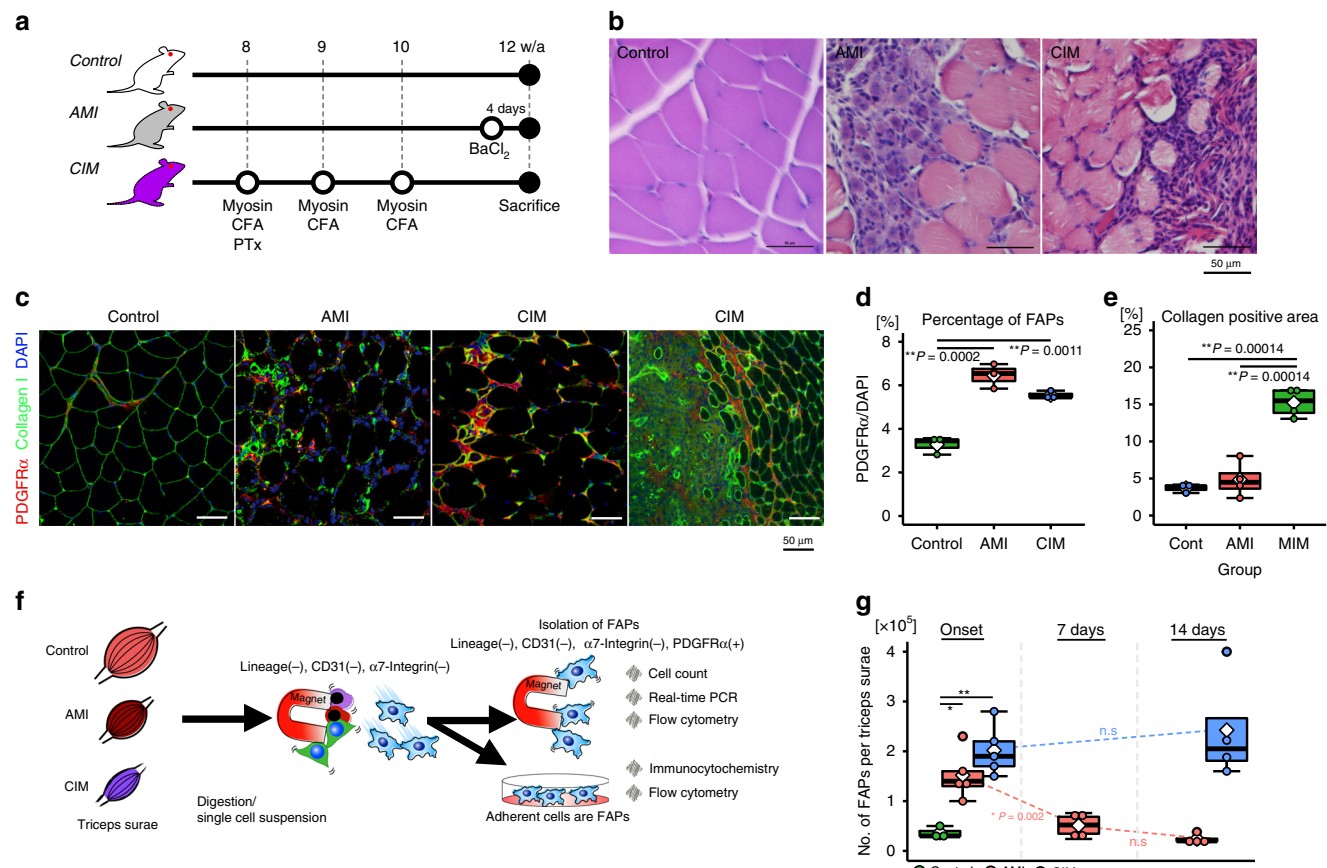

**Fig. 1 Fibro-adipogenic progenitor (FAP) accumulation is observed in chronic inflammatory myopathy model mice. a** Schematic diagram of the procedures used to establish the models of acute muscle injury (AMI) and chronic inflammatory myopathy (CIM). **b** Representative images of hematoxylin and eosin (H&E) staining of muscle from control, AMI, and CIM mice. Regenerating myofibers with central nuclei are observed in AMI muscle, while interstitial fibrosis and inflammatory cell infiltration are observed in CIM muscle. **c, d** Representative images of PDGFRα- and type I collagen-immunostained triceps surae (**c**), and quantification of the percentage of PDGFRα+ FAPs and fibrotic areas with collagen deposition in randomly chosen fields of view ($n = 3$ per group) (**d, e**). **f** Schematic diagram of the FAP isolation procedure. **g** The number of magnetic bead-sorted FAPs marked as lineage-negative (Lin−), CD31−, α7-integrin−, and PDGFRα+ in control, AMI, and CIM mice ($n = 3$ for control, $n = 5$ for AMI and CIM). Quantitative data are shown as means as well as medians with IQRs and 1.5 times the IQR, and are displayed by dot plot and box and whisker plot. *P* values were determined by one-way ANOVA adjusted by the Holm method (*$P < 0.05$, **$P < 0.001$). NS, not significant.

In addition, we investigated the activity of another cell senescence marker, senescence-associated β-galactosidase (SA-β-gal), and also γH2A.X expression, in FAPs. We found that SA-β-gal-positive FAPs were significantly increased in AMI-FAPs compared with control-FAPs and CIM-FAPs (Fig. 2b, c). γH2A.X, which results from phosphorylation of the Ser-139 residue of the histone variant H2A.X, is an early cellular response to the induction of DNA double-strand breaks[22]. Activation of a DNA damage response, including formation of DNA damage foci containing activated H2A.X either at uncapped telomeres or persistent DNA double-strand breaks, is known to be the major trigger of cell senescence[23], and therefore γH2A.X is used as a reliable quantitative indicator of senescent cells[24]. Expression of γH2A.X was increased in AMI-FAPs compared with CIM-FAPs (Fig. 2d, e).

We also examined the mRNA expression of *Tnfaip6* and *Il33*, which encode the cytokines TNF-α-stimulated gene-6 (TSG-6) and interleukin-33 (IL-33); the former promotes tissue regeneration via macrophages, while the latter promotes muscle-resident regulatory T cells and satellite cells[25,26]. The mRNA expression levels of T and *Il33* were decreased in CIM-FAPs compared with AMI-FAPs (Fig. 3f). Interestingly, p16^INK4A-expressing cells also showed high IL-33 expression (Fig. 3g), and mRNA expression of

*Cdkn2a* and *Trp53* in FAPs correlated strongly both with *Il33* and *Tnfaip6* expression, as well as with muscle function (Fig. 3h).

To test whether the differences in FAPs derived from AMI or CIM mice affected their ability to promote myogenesis, we conducted FAP-satellite cell transwell co-culture experiments (Fig. 3i). FAPs isolated from AMI mice, but not CIM mice, could potentiate satellite cells to form MyHC+ multinucleated myotubes (Fig. 3j–l).

To confirm whether FAPs derived from mdx mice, another model of chronic muscle inflammation, showed similar characteristics as CIM-FAPs, we conducted immunohistochemical analysis of PDGFRα and p16^INK4A in AMI, CIM, and mdx mice. Although PDGFRα+ FAPs in AMI mice expressed p16^INK4A, those in CIM and mdx mice did not, although p16^INK4A expression was detected in cells near PDGFRα+ FAPs (Supplementary Fig. 3A–C). The percentage of p16^INK4A+ FAPs was significantly higher in AMI mice than CIM and mdx mice (Supplementary Fig. 3D).

We also investigated the expression of β-klotho, a negative regulator of p16^INK4A, which is strongly expressed by FAPs in the skeletal muscle[27], because the upregulation of β-klotho expression was previously confirmed in mdx mice[28]. The mRNA expression of *Klb* in CIM-FAPs was significantly higher than in AMI-FAPs

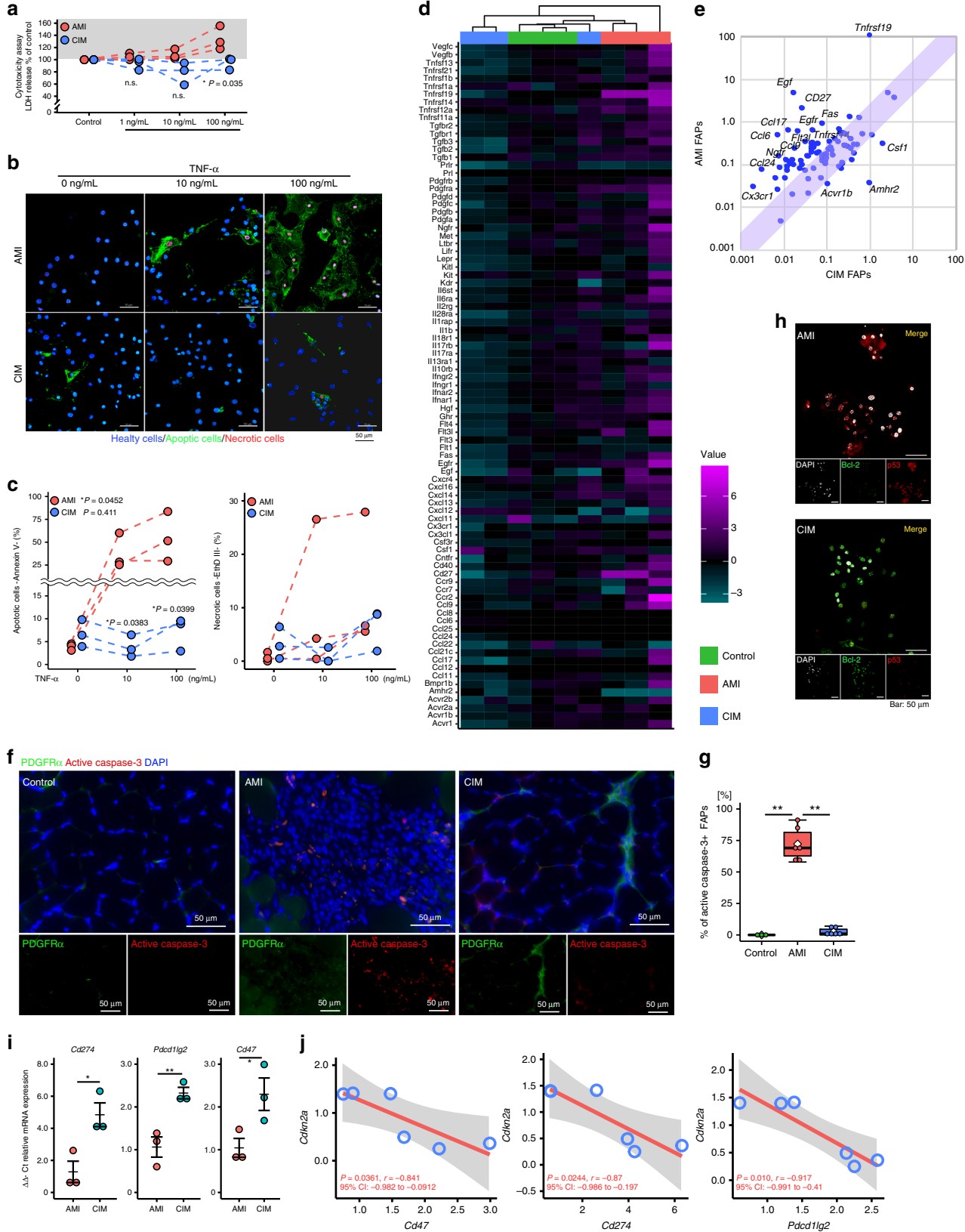

(Supplementary Fig. 3E), and *Klb* expression was strongly negatively correlated with the mRNA expression of *Cdkn2a* in FAPs (Supplementary Fig. 3F). Therefore, the senescent phenotypes of CIM-FAPs and mdx-FAPs may share underlying molecular mechanisms, specifically a low expression level of p16$^{INK4A}$ and a high expression level of β-klotho. We hypothesize

that FAP senescence is necessary to promote muscle regeneration and improve muscle function.

**Trp53 activation is required to promote muscle regeneration.** To investigate whether FAP senescence is necessary to promote

**Fig. 2 FAPs derived from chronic inflammatory myopathy model mice acquired features of apoptosis resistance. a** Cytotoxicity assay shows the LDH release level after stimulation with different concentrations of TNF-α in FAPs isolated from AMI and CIM ($n = 3$ per group). **b**, **c** Representative image of FITC-Annexin V (green) and ethidium homodimer III (red) after stimulation with 1, 10, and 100 ng/mL of TNF-α, and the quantified data. The number of Annexin V+, apoptotic FAPs increased in a dose-dependent fashion with the addition of TNF-α in AMI, but not in CIM mice ($n = 3$ per group). **d**, **e** Hierarchical clustering of differentially expressed cytokine–cytokine receptor gene expression was profiled by PrimerArray® analysis in control, AMI, and CIM mice ($n = 3$ per group) (**d**). Genes with higher expression are depicted in magenta, genes with lower expression are depicted in cyan, and genes with no difference are depicted in black (**d**). Scatterplots of gene expression changes in AMI-FAPs ($n = 3$) compared with CIM-FAPs ($n = 3$) (**e**). **f**, **g** Representative images of PDGFRα- and active caspase-3-immunostained triceps surae in control, AMI, and CIM mice (**f**), and quantification of the percentage of active caspase-3-positive FAPs in randomly chosen fields of view (**g**). **h** Representative confocal images of Bcl-2- and p53-immunostained FAPs isolated from AMI and CIM. **i** mRNA expression of *Cd274*, *Pdcd1lg2*, and *Cd47* in FAPs isolated from AMI and CIM mice ($n = 3$ per group). **j** Correlation of *Cd274*, *Pdcd1lg2*, and *Cd47* mRNA expression with *Cdkn2a* mRNA expression in FAPs isolated from AMI and CIM mice. Quantitative data for each specimen are shown in a dot plot. *P* values were determined by one-way ANOVA adjusted by the Holm method (*$P < 0.05$, **$P < 0.001$). NS, not significant.

muscle regeneration, we performed experiments involving the transplantation of *Trp53*(+/+) and *Trp53*(−/−) FAPs. *Trp53* (+/+) and *Trp53*(−/−) FAPs were isolated from *Trp53*(+/+) and *Trp53*(−/−) mice, respectively, using a magnetic cell isolation system. Then *Trp53*(+/+) or *Trp53*(−/−) FAPs were transplanted into the triceps surae of C57BL/6 mice, and 7 days after cell transplantation, barium chloride (BaCl₂) solution was injected into the triceps surae to induce AMI (Fig. 4a). At 10 and 20 days post injury (dpi), the muscles of mice transplanted with *Trp53*(−/−) FAPs showed a lower volume and wet weight than the muscles of mice transplanted with *Trp53*(+/+) FAPs (Fig. 4b, c). In histological analysis, mice transplanted with *Trp53*(+/+) FAPs showed regenerating muscle fibers at 10 dpi and complete regeneration at 20 dpi; in contrast, mice transplanted with *Trp53* (−/−) FAPs showed excessive cell accumulation at 10 dpi and interstitial fibrosis at 20 dpi (Fig. 4d). We also found that mice transplanted with *Trp53*(−/−) FAPs showed a smaller myofiber cross-sectional area than mice transplanted with *Trp53*(+/+) FAPs (Fig. 4e, d). Further, at 20 dpi, the number of FAPs was decreased in mice transplanted with *Trp53*(+/+) FAPs, but significantly higher number of FAPs was accumulated in mice transplanted with *Trp53*(−/−) FAPs (Fig. 4e, g). To better understand the mechanism whereby *Trp53*(−/−) FAPs mediated the impairment of muscle regeneration, we performed an in vitro study using a model involving H₂O₂ stimulation, because the production of reactive oxygen species in skeletal muscle has been shown to occur in muscle injury, including that induced by exercise[29,30]. *Trp53*(−/−) FAPs, but not *Trp53*(+/+) FAPs, showed an anti-apoptotic phenotype in response to high-dose H₂O₂ or TNFα stimulation (Fig. 4h, i). We confirmed that H₂O₂ stimulation induced the upregulation of *P21* and *Cdkn2a* mRNA expressions in *Trp53*(+/+) FAPs, but not in *Trp53*(−/−) FAPs (Fig. 4j). The mRNA expressions of *Cd274* and *Cd47* in *Trp53* (−/−) FAPs were higher than in *Trp53*(+/+) FAPs, and only *Trp53*(−/−) FAPs demonstrated increased *Cd274* expression following H₂O₂ stimulation (Fig. 4j). We also found that the expression of *Fst*, which promotes the differentiation of muscle stem cells, was increased after H₂O₂ stimulation in *Trp53*(+/+) FAPs, but was decreased in *Trp53*(−/−) FAPs (Fig. 4j). Next, to determine whether *Trp53*(+/+) FAPs promote immune clearance and muscle regeneration, and whether *Trp53*(−/−) FAPs impair immune clearance and muscle regeneration upon FAP damage, we performed co-culture experiments using RAW264.7 macrophages and C2C12 myoblasts. We found that untreated *Trp53*(+/+) and *Trp53*(−/−) FAPs showed no difference in the percentage of FAP phagocytosed by RAW264.7, but following H₂O₂ treatment this percentage was increased with *Trp53*(+/+) FAPs and decreased with *Trp53*(−/−) FAPs (Fig. 4k, l). C2C12 myoblast differentiation was greater when these cells were co-cultured with untreated *Trp53*(+/+) or *Trp53*(−/−) FAPs than when they were co-cultured without FAPs, but C2C12

differentiation was promoted by H₂O₂-treated *Trp53*(+/+) FAPs and inhibited by H₂O₂-treated *Trp53*(−/−) FAPs (Fig. 4m, o).

**Exercise promotes FAP senescence in normal mice**. Next, we investigated whether FAP senescence was related to muscle regeneration after exercise-induced damage. It is well known that exercise promotes muscle stem cell and FAP proliferation, and proper exercise is a powerful therapeutic tool for muscle disease[2]. On the other hand, exercise-induced damage can also cause exacerbation of inflammation and fibrosis in CIM[4,5]. To confirm that exercise-induced damage promotes senescence of FAPs, we analyzed FAPs from normal and CIM mice 24 h after downhill treadmill exercise (−20°, 17 m/min, 30 min; Fig. 5a). In normal mice, the number of FAPs was significantly increased by exercise-induced damage (Fig. 5b), whereas there was no increase in CIM mice (Fig. 5b). We found that the mRNA expression levels of *Cdkn2a* and *P21* in control-FAPs were increased by exercise (Fig. 5c), and *Tnfaip6* and *Il33* were also increased (Fig. 5c). However, the mRNA expression levels of *Cdkn2a* and *P21* in CIM-FAPs were decreased by exercise (Fig. 5c), and *Tgfb1* and *Acta1*, which are fibrosis-associated genes[8], were increased by exercise (Fig. 5c). The protein expression levels of p16^INK4A and p53 in FAPs were also decreased by exercise in CIM mice (Fig. 5d, e). Phospho-p38 mitogen-activated protein kinase (MAPK) and phospho-NF-κB p65 expression levels in control-FAPs were increased by exercise (Fig. 5f, g). When upregulated by exercise, MAPK and NF-κB are two major regulators of gene transcription and metabolism in response to mechanical stress in the skeletal muscle, and are essential for maintaining muscle homeostasis and preventing muscle atrophy[31]. On the other hand, in CIM-FAPs, phospho-NF-κB p65 expression was increased and phospho-p38 MAPK expression was decreased by exercise (Fig. 5f, g).

**Pro-senescent intervention in FAPs restores muscle function**. We investigated the therapeutic effects of a 14-day exercise intervention (−20°, 17 m/min, 30 min, daily; Fig. 6a). In control mice, the exercise intervention increased endurance capacity, hind limb grip strength, muscle cross-sectional area, and the number of regenerating muscle fibers (Fig. 6b–f). However, the exercise intervention showed a poor therapeutic effect in CIM mice (Fig. 6b–f). We next evaluated the levels of cell senescence in FAPs after the exercise intervention. The percentage of p16^INK4A+ FAPs and SPiDER-β-gal+ FAPs was increased in the control population, but not in CIM mice (Fig. 7a–c and Supplementary Fig. 4A, B). Next, we tried to promote FAP senescence by administering AICAR, as it is known to promote p53 and p16^INK4A [12,32]. We confirmed that p16^INK4A and p53 expression levels were upregulated by treatment with AICAR in vitro (Fig. 7d). The AICAR-treated FAPs also upregulated IL-33 and downregulated Bcl-2 expression (Fig. 7d), which was also

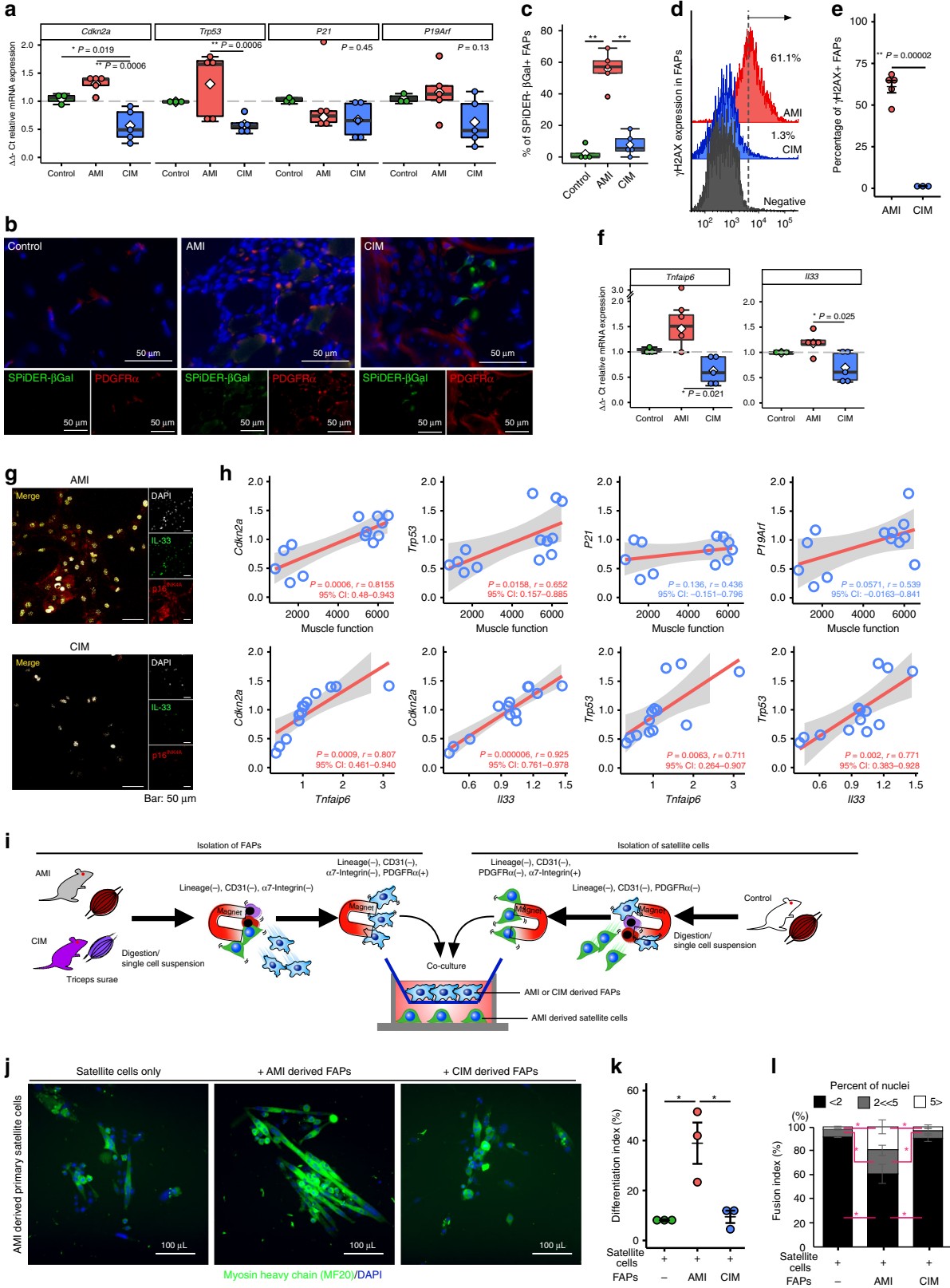

the case in AMI-FAPs (Fig. 3g). To investigate the therapeutic effect of a pro-senescent intervention, CIM mice were administered AICAR on a daily basis. This increased endurance capacity (Fig. 6b) and decreased the number of FAPs compared with baseline CIM mice (i.e., those that did not undergo the exercise intervention) (Fig. 7a, b). Further testing of the CIM exercise+

AICAR mice showed dramatically increased muscle function and muscle cross-sectional area, similar to the findings in control-sedentary mice, and regenerating muscle fibers also increased (Figs. 6b and 5f). Although the number of FAPs was not decreased, the percentage of p16[INK4A]+ FAPs and SPiDER-β-gal+ FAPs was increased compared with baseline CIM mice

**Fig. 3 FAPs acquire senescent features after acute muscle injury. a** Relative mRNA expression of senescence-related genes (*Cdkn2a*, *Trp53*, *P21*, and *P19Arf*) in FAPs from control (*n* = 3), AMI (*n* = 5), and CIM (*n* = 5) mice. **b, c** Representative images of SPiDER-β-gal- and PDGFRα-immunostained triceps surae (**b**), and quantification of the percentage of SPiDER-β-gal-positive FAPs in randomly chosen fields of view in control (*n* = 3), AMI (*n* = 5), and CIM (*n* = 5) (**c**) mice. **d, e** Representative γH2AX histograms from *n* = 3 replicates (**d**), and quantification of percentages of γH2AX+ FAPs (**e**). **f** Relative mRNA expression of *Tnfaip6* and *Il33* in FAPs from control (*n* = 3), AMI (*n* = 5), and CIM (*n* = 5) mice. **g** Representative confocal images of IL-33- and p16INK4A-immunostained FAPs isolated from AMI and CIM. **h** Correlation of *Cdkn2a*, *Trp53*, *P21*, and *P19Arf* mRNA expression with muscle function, and *Tnfaip6* and *Il33* mRNA expression in FAPs isolated from normal, AMI, and CIM mice. **i** Schematic diagram of satellite cells co-cultured in transwells with FAPs from AMI and CIM mice. **j–l** Representative images of MyHC (green) and DAPI (blue) after a 7-day transwell co-culture with or without FAPs (**j**), and quantification of myogenic differentiation of satellite cells assessed by the differentiation index and fusion index (**k, l**) (*n* = 3 per group). Quantitative data are shown as means as well as medians with IQRs and 1.5 times the IQR, and are displayed by dot plots and box and whisker plots, or shown as means ± SEM (dot plot). *P* values were determined by two-tailed Student's *t* test or one-way ANOVA adjusted by the Holm method (\**P* < 0.05, \*\**P* < 0.001).

(Fig. 7a–c and Supplementary Fig. 4a, b). To better understand FAP phenotype alterations, cytokine–cytokine receptor gene expression was profiled by PrimerArray® analysis. *K*-means clustering classified three of three CIM exercise mice into cluster 1, which was the same cluster containing the baseline CIM group (Fig. 7e). On the other hand, three of three CIM exercise+ AICAR mice were classified into cluster 2, which was the same cluster containing the AMI group and the control CIM exercise group (Fig. 7e). One of the three CIM AICAR-only mice (with no exercise) was classified into cluster 1, and the other two were classified into cluster 2 (Fig. 7e). Principal component analyses showed that CIM exercise+ AICAR mice and CIM AICAR-only mice were shifted to the right, and CIM exercise+ AICAR mice were located close to the cluster of the AMI and control-exercise mice, whereas CIM exercise-only mice were located further away from the cluster of the AMI and control-exercise mice (Fig. 7f). A shift to the right indicates the pro-inflammatory and pro-apoptotic phenotype, which is characterized by an increase in cytokines and cytokine receptors, especially CC-chemokine ligands and TNF receptor superfamilies.

## Discussion

Muscle inflammation leads to increased numbers of FAPs, which are a key regulator of muscle stem cells, the extracellular matrix, and debris clearance, but excessive accumulation of FAPs causes muscle degeneration, fibrosis, and severe functional deficits[7,8]. The differences in FAP phenotypes between the regenerative and degenerative states remain unknown.

Here we showed that AMI-FAPs increased after injury and subsequently declined to pre-damage levels, while CIM-FAPs continued to accumulate for 14 days after injury by acquiring cell death resistance. In addition, AMI-FAPs acquired features of senescence, whereas CIM-FAPs showed decreased expression of senescent factors. In response to a variety of stresses, mammalian cells undergo senescence. Although the crucial determinants of whether a cell responds to damage by undergoing cell survival or cell death (apoptosis) are the cell type and the nature and intensity of the damage, apoptosis is sometimes a response to acute/high stress, whereas cell survival can result from low/constitutive stress with low levels of senescent factors[11]. For example, p53, a senescence cell marker, is known to trigger apoptosis in response to cellular stress, but whether or not this occurs depends on stress and cell type, specific modulated genes, p53 levels, and transcriptional activity[33]. Low levels of p53 expression protect fibroblasts from the induction of apoptosis, and wild-type p53 expressed by embryonic fibroblasts induces apoptosis caused by ultraviolet-B-induced damage[34]; however, expression of the hypomorphic R172P p53 mutation abrogates p53-mediated apoptosis both by downregulating the pro-apoptotic factors PUMA and NOXA and inducing high expression levels of the pro-survival gene Bcl-2 while keeping cell cycle control mostly intact, and enhances inflammation and immunosuppression

relative to wild-type mice[35]. Similarly, a recent study showed that downregulation of Bcl-2 was associated with p16-mediated apoptosis in non-small-cell lung cancer cells[36]. Another study demonstrated that p16INK4A reactivation downregulated the expression of survivin, a crucial apoptotic regulator, and exhibited antitumor potency by downregulating AKt/survivin signaling in hepatocellular carcinoma cells[37]. In our study, expression levels of wild-type p53 and p16INK4A were increased in AMI-FAPs compared with CIM-FAPs, with concomitant upregulation of Bcl-2. During remodeling of stressed or injured tissues, senescent cells are detectable by neighboring cells, which promotes their elimination and rapid replacement by the homeostatic protective system. Senescent cells secrete hundreds of factors, including pro-inflammatory cytokines, chemokines, growth factors, and proteases; together, these properties characterize the senescence-associated secretory phenotype (SASP)[38]. Cells with the SASP can recruit and activate phagocytes and resolve fibrosis, whereas the absence of senescence in cancer cells and liver stellate cells induces excessive cell accumulation and fibrosis, respectively[39,40]. Our results showed that larger numbers of CD11b+ cells were recruited in AMI mice than in those with CIM mice, because of the high level of senescence in the former. The CD11b+ cell population, which include monocytes, macrophages, NK cells, and granulocytes, plays an important role in phagocytosis and muscle regeneration[41]. Further, AMI-FAPs expressed TNF receptor superfamilies, such as Fas, CD27, Tnfrsf14, and Tnfrsf1, which are known cell death receptors, and the TNF signaling pathway mediates cell apoptosis[14]. The cells also expressed CC-chemokine ligands and IL-33. IL-33 is expressed by endothelial cells, fibroblasts, and neuronal cells, and affects various cell types that express membrane ST2[42]. Additionally, IL-33 is a potent inducer of pro-inflammatory cytokines and chemokines by mast cells, and enhances the production of TNF-α by macrophages[42]. FAPs also express IL-33 after muscle injury, and IL-33 regulates T cells that promote muscle regeneration[25]. Thus, senescent FAPs express cell death receptors, and IL-33 that arises due to AMI causes activation of immune cell function and creates a state of regenerative inflammation[43]. On the other hand, our results showed that CIM-FAPs with downregulated p16INK4A and p53 acquired apoptosis resistance against TNF-α stimulation. Lemos et al. found that FAP-induced apoptosis of TNF-α-rich inflammatory macrophages is essential for muscle regeneration in AMI, but TGF-β-rich anti-inflammatory macrophages induced FAP survival and promoted muscle degeneration in chronic inflammation[10]. We further revealed another muscle degenerative mechanism that involved a different macrophage phenotype[10], and also showed that FAPs themselves were resistant to TNF-α-induced apoptosis. Additionally, we found that CIM-FAPs upregulated the expression of PD-L1, PD-L2, and CD47, which are negative regulators of p16INK4A. These three molecules are generally activated in variety of cancers and atherosclerosis[21,44], and they contribute to cell accumulation both by facilitating

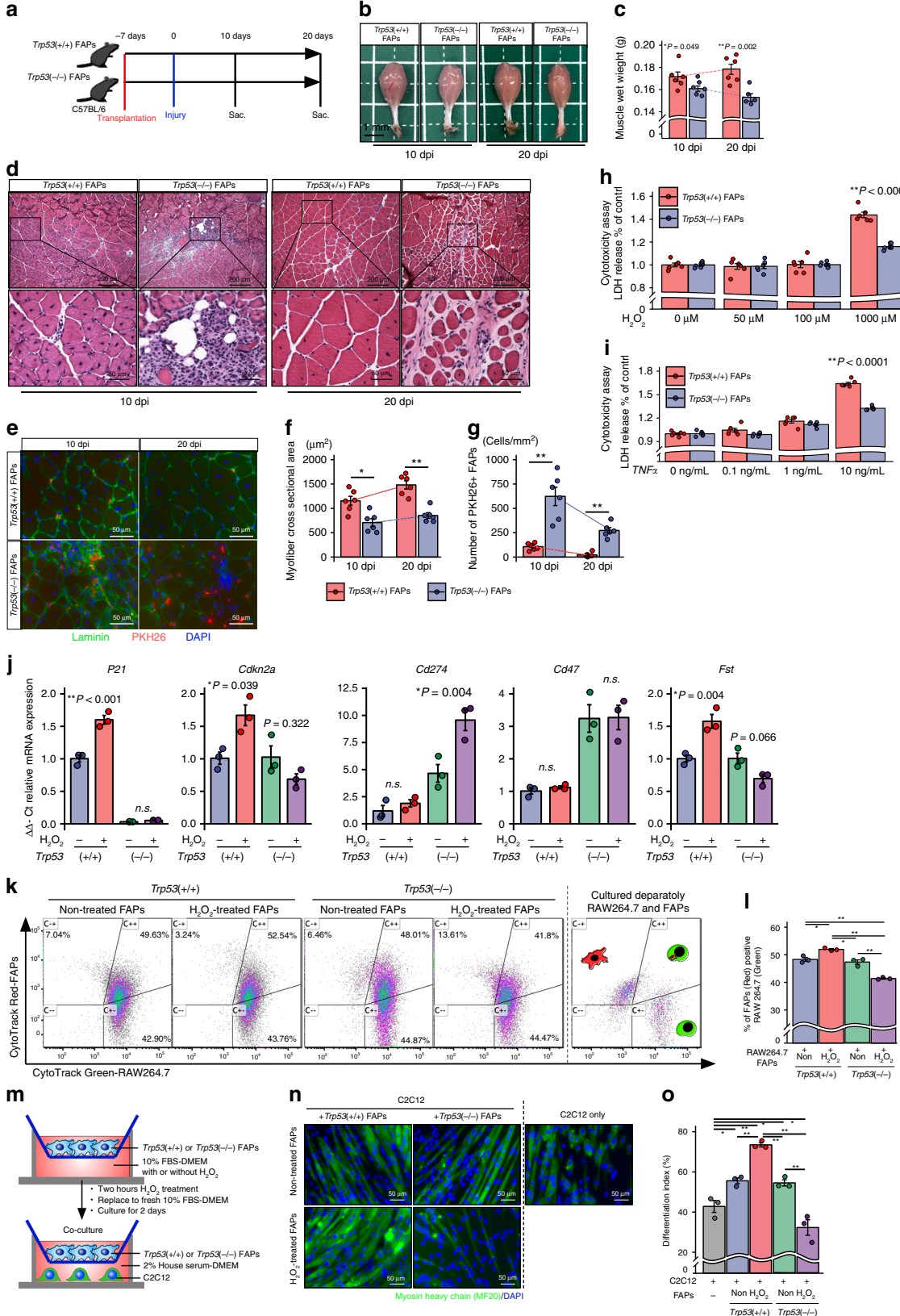

escape from the immune system and by inducing anti-phagocytic signaling, referred to as "don't find me" and "don't eat me" signals, respectively[20]. PD-L1 and PD-L2+ cells, often expressed in tumor cells and myofibroblasts, engage and cause dysfunction in PD-1+ cells, such as T cells and macrophages[21,45]. CD47 is

widely expressed on a majority of normal tissues and regulates phagocytosis, but its expression is upregulated in malignant cells[46]. CD47 engages signal-regulatory protein-α (SIRPα), which is mainly expressed on macrophages, resulting in SIRPα+ macrophages that are unable to phagocytose as well as

**Fig. 4 Trp53(−/−) FAP transplantation impairs muscle regeneration after acute muscle injury. a** Schematic diagram of the procedures for Trp53(+/+) or Trp53(−/−) FAP transplantation and $BaCl_2$-induced muscle injury. **b, c** Representative macro images of triceps surae at 10 and 20 days post injury (dpi) (**b**), and quantitative data of muscle wet weight (**c**) (n = 6 per group). **d** Representative images of H&E staining of muscle transplanted with Trp53(+/+) or Trp53(−/−) FAPs. **e–g** Representative images of PKH26-labeled FAPs and laminin-immunostained triceps surae (**e**), and quantitative data of the muscle cross-section area and the number of PKH26+-transplanted FAPs (**f, g**) (n = 3 per group). **h, i** The LDH release level after stimulation with different concentrations of $H_2O_2$ or TNF-α in Trp53(+/+) or Trp53(−/−) FAPs. **j** Relative mRNA expression of P21, Cdkn2a, Cd274, Cd47, and follistatin in Trp53 (+/+) or Trp53(−/−) FAPs with or without $H_2O_2$ stimulation (n = 3 per group). **k, l** Representative CytoTrack Green-labeled RAW264.7 and CytoTrack Red-labeled FAPs plots after co-culture with RAW264.7 and FAPs (**k**), and quantification of the percentages of double-positive cells (**l**) (n = 3 per group). **m** Schematic diagram of the procedure for the C2C12 and FAPs co-culture experiment. **n, o** Representative images of MyHC and DAPI staining of C2C12 (**n**), and the quantification of myogenic differentiation of C2C12 assessed by the differentiation index (**o**) (n = 3 per group). Quantitative data are shown as mean ± SE with dot plots. P values were determined by the paired t test or one-way ANOVA adjusted by the Holm method (*P < 0.05, **P < 0.001). NS, not significant.

---

inducing CD47+ cell accumulation[46]. PD-L1 and CD47 are negatively correlated with senescent factors, especially p16[INK4A] (Fig. 2h)[47,48], and therefore decreased senescence in CIM-FAPs can be considered to result from accumulation due to immune system escape and anti-phagocytic signaling. These results led to the hypothesis that pro-senescent interventions to induce apoptosis and SASP expression in FAPs will create a state of regenerative inflammation and thus comprise important therapeutic interventions in chronic myopathies.

Our study demonstrated that by regulating the senescence of FAPs, exercise-induced muscle damage could be effective for promoting muscle regeneration and attenuating chronic low-grade inflammation. Proper exercise seems to be a powerful therapeutic tool in muscle disease[2,49]. Exercise can promote senescence and activate the expression levels of the senescence-associated factors p38 MAPK and NF-κB in whole muscle tissue[31,50,51], and can also potently activate FAP proliferation and expression of muscle and vascular growth factors in normal and α7-integrin transgenic mice[49]. We demonstrated that exercise promoted FAP senescence in normal mice, but exercise in CIM mice decreased the expression of senescent factors to levels seen in sedentary CIM mice. In addition, exercise in CIM mice decreased the expression of phospho-p38 MAPK and markedly increased that of phospho-NF-κB p65. Both p38 MAPK and NF-κB are involved in cellular stress-related signaling, and p38 MAPK mediates apoptosis by regulating Bim, a member 2 of the pro-apoptotic Bcl-2 family of proteins, and by promoting the nuclear localization of FOXO proteins[52–54], while FOXO protein suppression by NF-κB promotes the acquisition of apoptosis resistance[55]. It is well known that excessive exercise induces fibrosis and degeneration by overactivation of NF-κB and TGF-β[4,5]. Furthermore, p16[INK4A] is a suppressor of NF-κB[56,57], and therefore by increasing the level of activated NF-κB, exercise might also regulate the expression of p16[INK4A]. Our results indicated that exercise in CIM mice had a poor therapeutic effect because it induced overactivation of NF-κB and TGF-β, as well as concurrent underactivation of p38 MAPK and p16[INK4A]. Hence, we tested whether the combined application of exercise and pro-senescent intervention targeting FAPs had a positive effect in mice with chronic myopathy.

Although the detailed mechanism of exercise-induced senescence is not well known, we assume it is related to AMPK activation. AMPK is involved in cellular energy homeostasis, and regulates cell growth and apoptosis[32]. AMPK activation occurs in response to metabolic stressors, such as hypoxia, ischemia, heat shock, and exercise[58], and activated AMPK promotes p16[INK4A] expression in fibroblasts[12]. Thus, we used AICAR, a cell-permeable APMK activator, to investigate whether AMPK activation would promote FAPs senescence and/or have a marked therapeutic effect in CIM mice. AICAR had some therapeutic effect, for example, improving exercise tolerance and decreasing

the number of FAPs. The decline in FAPs might have been caused by AMPK activation, as this process is known to be important for the efficient phagocytic removal of debris after tissue damage[59]. AICAR treatment in exercised CIM mice showed dramatic therapeutic effects, for example, increasing myofiber cross-sectional area, regenerating myofibers, and increasing p16[INK4A] expression in FAPs. The FAPs derived from exercised CIM mice treated with AICAR showed a similar expression pattern of cytokines and cytokine receptors as FAPs from AMI and exercised control mice. We found that AMI-FAPs had a pro-inflammatory and pro-apoptotic phenotype that promoted muscle regeneration, and the anti-apoptotic and pro-fibrotic phenotype of CIM-FAPs was altered by exercise with AICAR administration. AMPK activation can be achieved using AICAR and exercise, but we detected different therapeutic effects under each of the following conditions: exercise alone, AICAR treatment alone, and exercise with AICAR treatment. It has been shown that AICR activates AMPK by acting as an AMP mimetic that binds to AMP binding sites on intracellular proteins, thus leading to metabolic stress[32]. Exercise-induced AMPK activation also causes metabolic stress, but it simultaneously increases mechanical stress[60]. The reasons for the differences in therapeutic effects between AICAR and exercise are that the mechanical stress is only caused by exercise. Mechanical stress regulates cell fate via adherence junctions, the cytoskeleton, and integrins, and excessive mechanical stress causes tissue fibrosis by producing the pro-fibrotic factors TGF-β and CTGF, as well as the anti-apoptotic factor Bcl-2[61]. Metabolic stress interacts with mechanical stress, and metabolic stress due to activated AMPK can affect cell shape and cytoskeletal dynamics, which contribute to mechanosensitivity[62]; as a result, metabolic stress affects the sensitivity to mechanical stress. Muscle tissues are stiffer in dystrophic mice (mdx mice) than in normal mice, and this stiffness is heterogeneous; furthermore, fibroblasts in mdx mice have higher mechanosensitivity than those in normal mice[63]. The fibroblasts in stiff muscle can easily experience higher mechanical stress than fibroblasts in more compliant muscle when subjected to the same mechanical stress. Furthermore, AMPK activity is depressed in mdx mice[64]; that is, the mechanical stress sensitivity is higher and the metabolic stress sensitivity is lower in fibroblasts in chronic myopathy. Thus, our intervention that combines exercise with AICAR administration has a better therapeutic effect than exercise alone as a result of increasing metabolic stress and decreasing mechanical stress.

One of the most intriguing results of our study is that subjecting FAPs to the pro-senescent intervention consisting of exercise and AICAR treatment provided powerful therapeutic effects in a murine CIM model. However, this experimental model is an autoimmune myositis model, and the symptoms are therefore not as severe as in a mouse model of muscular dystrophy, but our results from immunohistochemistry of FAP

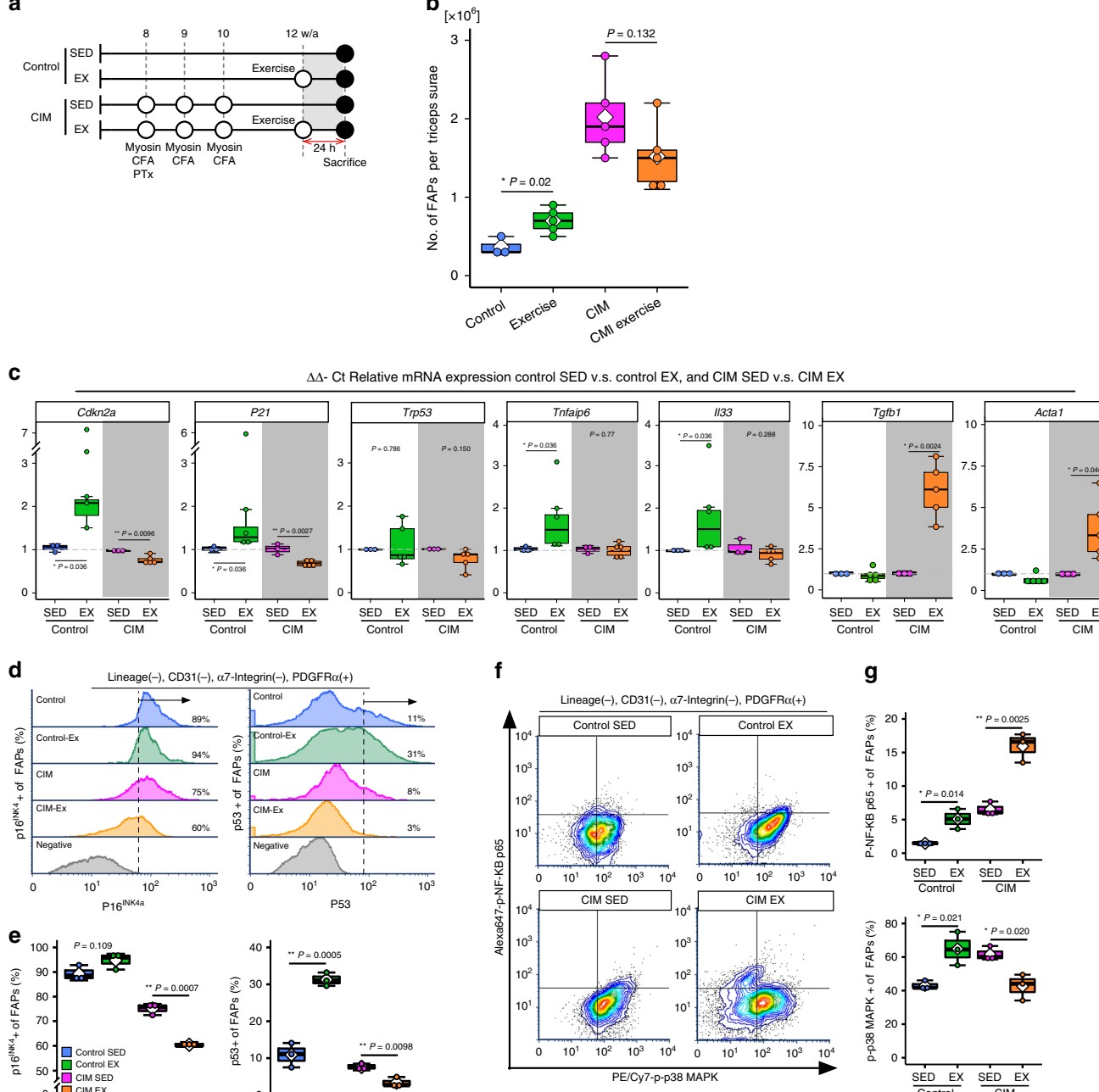

**Fig. 5 Exercise-induced senescence is observed in normal mice. a** Protocols for the CIM model and exercise intervention. **b** The number of FAPs in sedentary ($n = 3$) vs. exercise control mice ($n = 5$), and sedentary ($n = 5$) vs. exercise CIM mice ($n = 5$). **c** Relative mRNA expression of senescence-related genes (*Cdkn2a*, *Trp53*, and *P21*), pro-regenerative genes (*Tnfaip6* and *Il33*), and pro-fibrotic genes (*Tgfb1* and *Acta1*) in FAPs from sedentary control ($n = 3$), exercise control ($n = 5$), sedentary CIM ($n = 3$), and exercise CIM ($n = 5$) mice. **d, g** Flow cytometric analysis of p16$^{INK4A}$, p53, phospho-p38 MAPK, and phospho-p65 NF-κB in FAPs from sedentary control, exercise control, sedentary CIM, and exercise CIM mice ($n = 3$ in each group). **d** Representative p16$^{INK4A}$ and p53 histograms from $n = 3$ replicates, and **e** quantification of the percentage of p16$^{INK4A}$+ or p53+ FAPs. **f** Representative phospho-p38 MAPK and phospho-p65 NF-κB contour plot from $n = 3$ replicates, and **g** quantification of the percentages of phospho-p38 MAPK+ and phospho-p65 NF-κB+ FAPs. Quantitative data are shown as means as well as medians with IQRs and 1.5 times the IQR, and are displayed by dot plots and box and whisker plots. $P$ values were determined by the two-tailed Student's $t$ test (*$P < 0.05$, **$P < 0.001$).

senescence, FAP-satellite cell co-culture, and β-klotho expression suggest that FAPs have a similar phenotype in CIM and mdx mice. The concurrent application of AICAR and exercise requires further investigation because disease severity and the intensity of exercise could affect FAP senescence. In addition, it is important to know the differences in regeneration processes resulting from AICAR, exercise, and their combination so as to improve the therapeutic effects. Furthermore, pro-senescent interventions are

thought to be useful as a therapy for cancer and fibrosis, but not for tissue regeneration, especially in the case of somatic stem cells. While senescent non-myogenic cells participate in muscle regeneration, their aging-related expression of senescent factors inhibits muscle stem cell self-renewal and reduces regeneration potential[65,66]. In our model, it is unknown if muscle stem cells acquired features of senescence as a result of AICAR and exercise, and if so, if this played some negative role; however, AICAR and

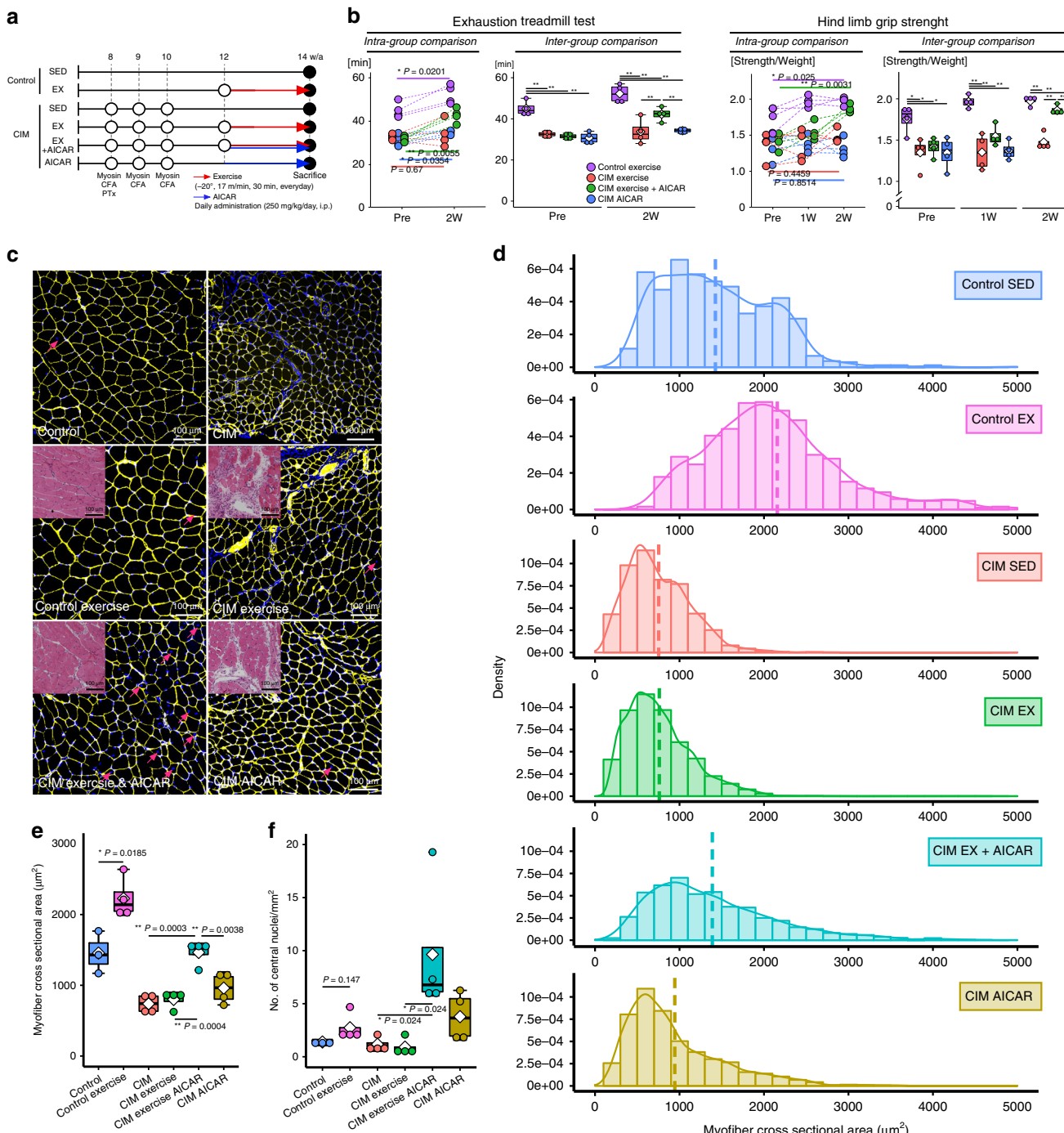

**Fig. 6 Muscle strength restored in CIM mice by combined exercise and AICAR treatment. a** Protocol for 2-week therapeutic intervention with exercise and AICAR treatment in control and CIM mice. **b** Results of the exhaustion treadmill test and hind limb grip strength test are shown as intra- (left panel) and inter-group (right panel) comparisons (*n* = 3 per group). **c–f** Changes in muscle cross-sectional area and number of regenerating muscle fibers in normal and CIM mice following exercise and/or AICAR treatment. **c** Representative images of H&E- and laminin-immunostained triceps surae (yellow) and nuclei (blue). Arrows indicate the central nuclei of regenerating muscle fibers. **d** The distribution of cross-sectional triceps surae fiber areas, with the mean area indicated by the dotted line. **e, f** Quantitative data of muscle cross-section area and the number of regenerating muscle fibers (*n* = 3 for control, *n* = 4 for exercise and CIM). Data are shown as a dot plot for each specimen, and as means as well as medians with IQRs and 1.5 times the IQR by dot plots and box and whisker plots. *P* values were determined by the paired *t* test or one-way ANOVA adjusted by the Holm method (*\*P* < 0.05, \*\**P* < 0.001).

exercise have positive effects on muscle stem cells via AMPK activation[67]. Further, cell senescence is necessary for maintaining tissue homeostasis and repair as a result of activation of immune cells such as monocytes/macrophages and T cells[68], and these immune cells strongly interact with FAP activity[10,25]. Further research is needed concerning the relationship between cell

senescence and cell metabolism on the one hand, and muscle stem cells, FAPs, and immune cells on the other.

In conclusion, we report a mechanism whereby FAP acquisition of senescent features promotes muscle regeneration by inducing apoptosis and SASP expression, creating a state of regenerative inflammation. Failure of FAPs to enter a senescent

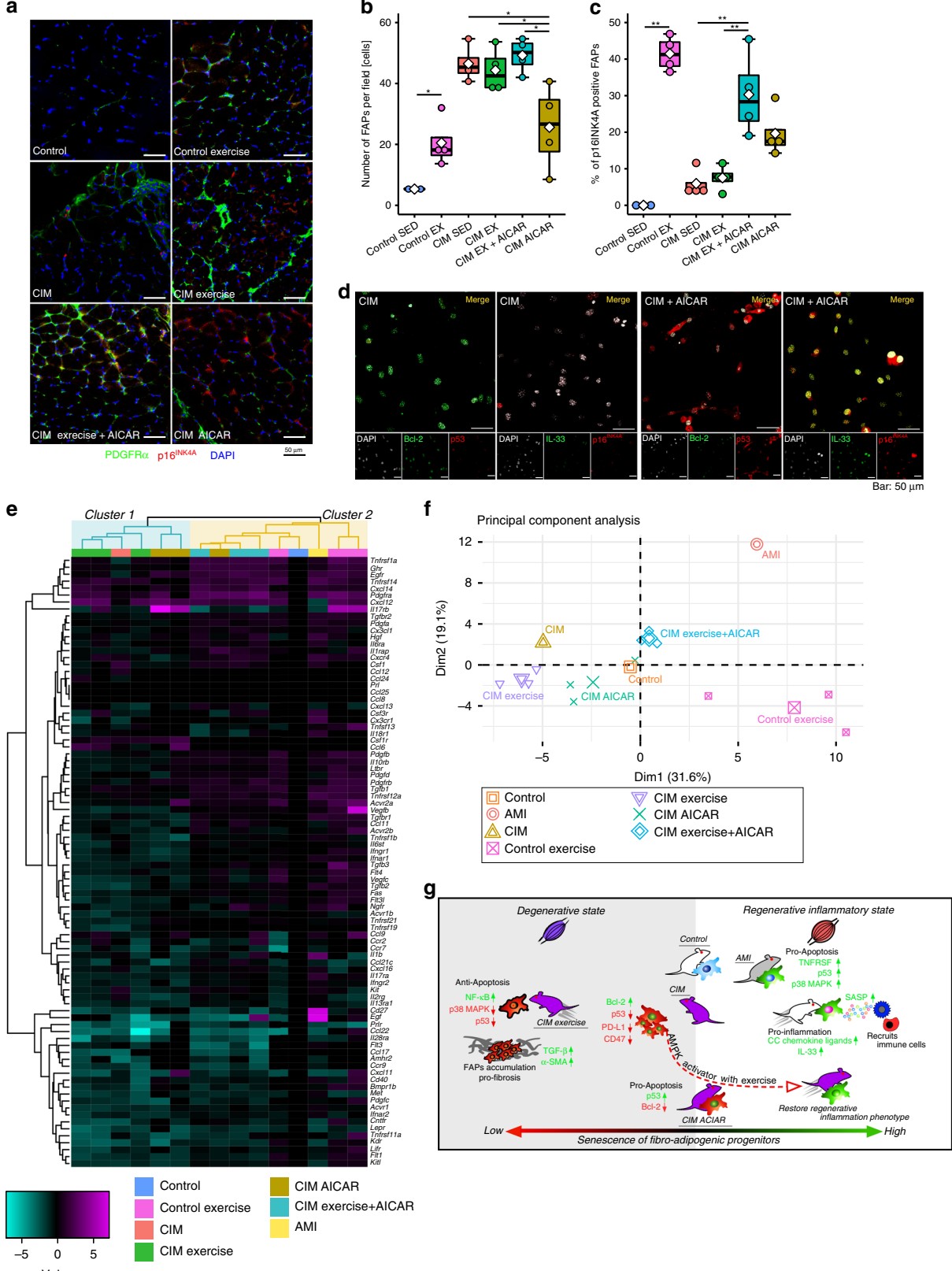

state after muscle damage, including that resulting from exercise, leads to muscle degeneration and FAP accumulation due to these cells' acquisition of an anti-apoptotic and pro-fibrotic phenotype (Fig. 7g). Subjecting FAPs to the pro-senescent intervention of exercise and pharmacological AMPK activation may serve as a therapeutic strategy for CIM.

## Methods

**Animals.** The Committee of the Animal Experimentation Center of the Sapporo Medical University School of Medicine approved all animal protocols. Mice were maintained in an enclosed, pathogen-free facility. Female BALC/c and C57BL/6 mice (age >6 weeks; Sankyo Lab Service, Tokyo, Japan) were used in all experiments. *B6.Cg-Trp53 < tm1Sia > /Rbrc* mice (*C57BL−p53+/−*) (Accession No. CDB 0001K) were provided by the RIKEN BioResource Center (Ibaragi, Japan)[69].

**Fig. 7 Exercise with AICAR treatment resulted in a pro-inflammatory and pro-apoptotic FAP phenotype that promotes muscle regeneration. a–c** Representative images of PDGFRα- and p16[INK4A]-immunostained triceps surae (green and red, respectively) (**a**), and quantitative data of the number of FAPs (**b**) and the percentage of p16[INK4A]+ FAPs (**c**) in randomly chosen fields of view (n = 3 for control, n = 4 for exercise and CIM). **d** Representative confocal images of Bcl-2-, p53-, IL-33-, and p16[INK4A]-immunostained FAPs isolated from CIM with or without AICAR treatment in vitro. **e** Hierarchical clustering of differentially expressed cytokine–cytokine receptor gene expression was profiled by PrimerArray® analysis in control, AMI, and CIM mice (the average value is that shown in Fig. 2d; n = 3 per group), and control exercise, CIM exercise, CIM AICAR, and CIM exercise with AICAR (n = 3 per group). Genes with higher expression are depicted in magenta, genes with lower expression are depicted in cyan, and genes with no difference are depicted in black. **f** Principal component analysis (PCA) of FAPs isolated from control, control exercise, CIM, CIM exercise, CIM AICAR, and CIM exercise with AICAR. **g** Proposed mechanism whereby exercise-induced FAP senescence promotes muscle regeneration. Data are shown as a dot plot for each specimen, and as means as well as medians with IQRs and 1.5 times the IQR by dot plots and box and whisker plots. P values were determined by the one-way ANOVA adjusted by the Holm method (*P < 0.05, **P < 0.001).

Genotypes were confirmed by PCR analysis. The primer sequences are presented in Supplementary Table S3. Male $Trp53(+/+)$ and $Trp53(-/-)$ mice were used in all experiments.

**Muscle injury model.** Mice were anaesthetized using isoflurane. To create the AMI model, 50 µL of 1.2% $BaCl_2$ (Sigma-Aldrich, Missouri, USA) in sterile distilled water was injected into the triceps surae of mice. We created the CIM model for use as an experimental autoimmune myopathy model by immunizing mice with partially purified myosin, including myosin-binding protein C. Myosin was partially purified according to the previously reported method, with few modifications[13]. Skeletal muscle from 8–10-week-old BALB/c mice was kept at −80 °C and then thawed and minced. Thirty grams of minced muscle was washed four times with 0.03 M KCl/0.15 M sodium phosphate buffer (PB; pH 7.5), 1 mM EDTA, and 1 mM dithiothreitol (DTT). Ninety milliliters of chilled 0.3 M KCl/0.15 M PB, 5 mM $MgCl_2$, 5 mM ATP, 1 mM EDTA, and 1 mM DTT was added to 30 g muscle and incubated on ice for 45 min under constant agitation to extract myosin. The homogenate was centrifuged at $2200 \times g$ for 30 min at 4 °C and the supernatant was collected and filtered. To precipitate the myosin, the filtrate was diluted 15 times with chilled ultrapure water. The precipitate was collected by centrifugation at $10,000 \times g$ for 10 min at 4 °C, dissolved in 0.5 M KCl, and stored at −80 °C. Eight-week-old BALB/c mice were immunized three times each, at 1-week intervals, with 200 µL emulsion containing 1 mg myosin and 100 µg complete Freund's adjuvant (Chondrex Inc., Washington, USA). This emulsion was injected bilaterally into the hock (first immunization), tail base (second immunization), and flanks (third immunization). Pertussis toxin (List Biological Laboratories Inc., California, USA) was injected intraperitoneally (500 ng in 100 µL saline) 1 h after the first immunization. We confirmed the elevation of serum antibody titers against myosin by enzyme-linked immunosorbent assay and by immunohistochemical staining of CD8+ T cells and major histocompatibility complex class I (Supplementary Fig. 1A, B).

**Exercise studies.** For the acute exercise study, 13-week-old control and CIM mice were acclimated to and trained on a −20° downhill treadmill for 30 min at 17 m/min. Speed on the treadmill was gradually increased from 10 to 17 m/min during a 2-min warm-up period. The muscle tissues were harvested 24 h after exercise. For long-term exercise, control and CIM mice were trained on a −20° downhill treadmill for 2 weeks every day for 30 min at 17 m/min. CIM mice were divided into four groups, including (1) without daily exercise (sedentary; SED), (2) with daily exercise (EX), (3) with AICAR treatment (250 mg/kg per day, intraperitoneally (i.p.)) and exercise, and (4) with AICAR treatment (250 mg/kg per day, i.p.) but without exercise. The muscle tissues were harvested 24 h after the final exercise session.

**Cell isolation.** Triceps surae were carefully dissected to remove attached tendons, nerves, blood vessels, and fat tissue. Muscles were dissociated mechanically and digested in 0.2% collagenase type 2 (Worthington Biochemical Corporation, New Jersey, USA) and 2.5 mM $CaCl_2$ in phosphate-buffered saline (PBS) for 60 min at 37 °C. Digested muscles were passed through an 18-gauge needle several times and further digested for 30 min at 37 °C. Muscle slurries were filtered through a 100-µm cell strainer (EASYstrainer™ Cell; Greiner Bio-One, Kremsmuenster, Austria) and through a 40-µm cell strainer (Greiner Bio-One). Erythrocytes were eliminated by treating the cells with RBC lysis solution (Qiagen, Hilden, Germany).

The Miltenyi MACS purification system was used to isolate FAPs for gene expression analysis by real-time PCR. Briefly, the cells isolated from digested muscles were resuspended in MACS buffer consisting of PBS with 2% fetal bovine serum (FBS) and 2 mM EDTA, and incubated with FcR blocking reagent (Miltenyi Biotec, Bergisch Gladbach, Germany) for 10 min at 4 °C, followed by incubation with biotin-conjugated Lineage Cell Detection Cocktail (Miltenyi Biotec), anti-CD31 (BioLegend, California, USA), and α7-integrin (Miltenyi Biotec) for 15 min at 4 °C. Subsequently, cells were washed once with MACS buffer, and incubated with anti-biotin microbeads (Miltenyi Biotec). Antibody–microbead cellular complexes were passed through a magnetic LD column and the flow through the fraction was collected (Lin−, CD31−, α7-integrin− cells). Then, Lin−/CD31−/α7-

integrin− cells were incubated with rabbit anti-PDGFRα antibody for 15 min at 4 °C, and then with anti-biotin microbeads (Miltenyi Biotec). Antibody–microbead cellular complexes were passed through a magnetic MS column (Miltenyi Biotec), and cells attached to the column were collected and defined as FAPs (Lin−/CD31 −/α7-integrin−, and PDGFRα+ cells).

FAPs for culture were isolated by negative selection using an EasySep™ Magnet (STEMCELL Technologies Inc.). Briefly, the cells isolated from digested muscles were resuspended in 2% FBS and 2 mM EDTA in PBS, and incubated with rat serum and FcR blocking reagent (Miltenyi Biotec), followed with incubation with biotin-conjugated Lineage Cell Detection Cocktail (Miltenyi Biotec), anti-CD31 (BioLegend), and α7-integrin (Miltenyi Biotec) for 10 min at room temperature (RT). Subsequently, cells were incubated with EasySep Mouse Streptavidin RapidSpheres (STEMCELL Technologies Inc.) and Lin−/CD31−/α7-integrin− cells were isolated according to the manufacturer's instructions. The isolated cells were cultured in non-coated plastic plates in Dulbecco's modified Eagle's medium (DMEM) supplemented with 20% FBS and 1% penicillin/streptomycin, and the adherent cells expressing PDGFRα and Sca-1 were defined as FAPs (Supplementary Fig. 5).

**FAP transplantation model.** FAPs were isolated from the muscles of $Trp53(+/+)$ or $Trp53(-/-)$ mice by an EasySep™ Magnet. Isolated FAPs were incubated in non-coated plastic plates in DMEM supplemented with 20% FBS and 1% penicillin/streptomycin. Passage 2 or 3 FAPs were used for intramuscular transplantation. Fourteen-week-old wild-type C57BL/6J mice were used for recipient mice, and $Trp53(+/+)$ or $Trp53(-/-)$ FAPs were injected into the triceps surae (cell numbers: $8 \times 10^4$ cells; volume: 25 µL). Seven days after cell transplantation, 1.2% $BaCl_2$ solution was injected into the triceps surae to induce AMI as mentioned above.

**Co-culture of satellite cells and FAPs.** Cell culture inserts with 0.4-µm pores and 12-well culture plates (Corning) were used for transwell co-culture. Freshly isolated Lin−/CD31−/α7-integrin−/PDGFRα+ FAPs were plated on the upper inserts, and Lin−/CD31−/ PDGFRα−/α7-integrin+ satellite cells were plated on the bottom of the culture plate and cultured for 7 days. We calculated the differentiation index as the number of nuclei within MyHC+ myotubes with ≥2 nuclei as a percentage of the total nuclei, and the fusion index was measured as the percentage of MyHC− or MyHC+ myotubes that were mononucleated (<2 nuclei), the percentage of MyHC+ myotubes between two and five nuclei, and the percentage the MyHC+ myotubes with more than five nuclei[9].

**Co-culture of C2C12 and FAPs.** Cell culture inserts with 0.4-µm pores and 12-well culture plates (Corning) were used for transwell co-culture. Lin−/CD31−/α7-integrin− FAPs were cultured until passage 2, and plated on the upper inserts at passage 3 for the co-culture experiment. Twenty-four hours after incubation on the upper inserts, FAPs were treated with $H_2O_2$ for 2 h and the medium was then replaced with fresh DMEM with 10% FBS. C2C12 cells were grown independently from FAPs in DMEM with 10% FBS for 72 h, and then upper inserts themselves with FAPs were transferred into the C2C12-cultured plates. To promote C2C12 differentiation, DMEM with 2% horse serum (Sigma) was used for an additional 72-h co-culture. After 72 h, cells were fixed with 4% paraformaldehyde and immunostained with MyHC antibody. We calculated the differentiation index as the number of nuclei within MyHC+ myotubes with ≥2 nuclei as a percentage of the total nuclei. C2C12 myoblasts were purchased from European Collection of Authenticated Cell Cultures (ECACC). Cells were tested for mycoplasma using the e-Myco™ Mycoplasma PCR Detection Kit (iNtRON Biotechnology Inc., Seong-nam-si, South Korea).

**Flow cytometry analysis.** Magnetic isolated Lin−/CD31−/α7-integrin− cells (see above) were stained with different antibodies and analyzed by flow cytometry. The isolated cells were stained with Zombie Violet™ Dye (BioLegend) in PBS at a 1:100 dilution for 15 min, washed, and fixed and permeabilized with the PerFix-nc Kit (Beckman Coulter) for intercellular cytokines, or the PerFix-EXPOSE Phospho

Epitope Exposure Kit (Beckman Coulter) for the detection of phosphorylated intracellular antigens, all according to the manufacturer's instructions. Concentrations and sources of antibodies that were used are listed in Supplementary Table 1. Flow cytometry analysis was performed using FACSCanto™ II (BD Biosciences) equipped with 405-, 488-, and 633-nm lasers. Data collection and analysis were conducted using FACSDiva™ (BD Biosciences) and FCS Express 6 (De Novo software). The gating schemes of FAPs are provided in the dot and contour plot in Supplementary Fig. 6.

**RNA extraction, quantitative real-time PCR analysis**. Total RNA was isolated from MACS-isolated FAPs (Lin−/CD31−/α7-integrin− and PDGFRα+ cells) using Tri Reagent® (Molecular Research Center Inc., Cincinnati, OH), and the RNA was reverse transcribed into cDNA using an Omniscript RT Kit (205113; Qiagen, Hilden, Germany), iScript cDNA Synthesis Kit (1708891; Bio-Rad, Hercules, CA, USA), or PrimeScript RT Master Mix (RR036A; Takara Bio Inc., Shiga, Japan). Quantitative PCR was performed with Power SYBR® Green Master Mix (4368702; Applied Biosystems, Foster City, CA, USA) using the Applied Biosystems 7500 Real-Time PCR System (Applied Biosystems) under the following cycling conditions: 50 °C for 2 min and 95 °C for 10 min, followed by 40 cycles of amplification (95 °C for 15 s and 60 °C for 1 min). Expression levels were normalized to 18S RNA and GAPDH (glyceraldehyde 3-phosphate dehydrogenase). Specific primer sequences used for PCR are listed in Supplementary Table 2. The ΔΔCt method was used to compare data. The PrimerArray® Cytokine–Cytokine Receptor Interaction Kit (Mouse) (PN001; Takara Bio Inc.) was used with TB Green Premix Ex Taq II (Tli RNaseH Plus; Takara Bio Inc.) with the Applied Biosystems 7500 Real-Time PCR System (Applied Biosystems) under the following cycling conditions: 95 °C for 30 s, followed by 40 cycles of amplification (95 °C for 5 s and 64 °C for 34 s). Expression levels were calculated by the PrimerArray Analysis Tool Ver. 2.2 (Takara Bio Inc.).

**Histology and immunofluorescence**. Triceps surae were frozen in isopentane cooled with liquid nitrogen, and stored at −80 °C until analysis. Quadriceps from 4-month-old male C57BL/10ScSn-Dmdmdx/J mice (mdx) were fixed on 4% paraformaldehyde overnight. The following day, the tissue was transferred to sucrose in PB and incubated overnight, and OCT-embedded tissues were frozen in liquid nitrogen and stored at −80 °C until use. Cryosections (8-μm thickness) were cut by cryostat and the sections were air dried, fixed on 4% paraformaldehyde, washed in PBS, and immunostained or stained with hematoxylin and eosin or SPiDER-β-gal. For immunostaining, the tissue sections were incubated in 0.01 M PBS containing 0.3% Triton-X (PBS-T) and treated with 2% bovine serum albumin (BSA) for 60 min at RT. After washing with 0.01 M PBS-T, the sections were incubated with primary antibodies at 4 °C overnight, followed by secondary staining. Concentrations and sources of antibodies that were used are listed in Supplementary Table 1. Cells were cultured on an 8-well chamber slide (Thermo Scientific™ Nunc™ Lab-Tek™; Thermo Fisher Scientific, Waltham, MA). Cultured cells were fixed with 4% paraformaldehyde for 15 min at RT and then incubated in PBS-T and treated with 2% BSA for 60 min at RT. After washing with PBS-T, the cells were incubated with primary antibodies at 4 °C overnight, followed by secondary staining. Concentrations and sources of antibodies that were used are listed in Supplementary Table 1. Secondary antibodies were coupled to Cyanine Dyes 3, Alexa-488, or Alexa-647 fluorochromes, and nuclei were stained using 4′,6-diamidino-2-phenylindole (DAPI) (1:1000; D523, Dojindo, Kumamoto, Japan). After washing, tissue sections were mounted with VECTASHIELD (Vector Laboratories, California, USA). For the SPiDER-β-gal stain, the tissue sections were incubated in 0.01 M PBS containing 0.3% Triton-X (PBS-T) and incubated in a 20 μM SPiDER-β-gal (Dojindo) solution for 60 min at 37 °C. After washing of tissue sections, nuclei were stained using DAPI (Dojindo), and tissue sections were mounted with VECTA-SHIELD (Vector Laboratories). Sections were observed with a confocal laser scanning microscope (Nikon/A1; Nikon, Tokyo, Japan) or fluorescence microscope BZ-X700 (Keyence Corp., Osaka, Japan).

**Cell viability and apoptosis assay**. Freshly isolated FAPs were cultured on 8-well chamber slides or 96-well culture plates in DMEM supplemented with 20% FBS and 1% penicillin/streptomycin for 2 days, and then culture medium was replaced with DMEM supplemented with 2% FBS and 1% penicillin/streptomycin. To induce cell death, cells were treated with or without TNF-α (1, 10, and 100 ng/mL; BioLegend) and/or AICAR (0.5 mM; 10010241, Cayman Chemical) as per the experimental design. After 2 days culture, a cytotoxicity LDH assay was performed using the Cytotoxicity LDH Assay-WST (Dojindo), and apoptotic and necrotic cell detection assays were performed by staining with FITC-Annexin V, Ethidium Homodimer III, and Hoechst33342 (Apoptotic/Necrotic/Healthy Cells Detection Kit; PromoKine, Germany) according to the manufacturer's instructions.

**H₂O₂ stimulation of FAPs**. FAPs were seeded on a culture plate and incubated in DMEM with 20% FBS for 48 h. FAPs were treated with $H_2O_2$ (50, 100, and 1,000 μM; Junsei Chemical Co., Ltd.) for 2 h. Then, the culture medium containing $H_2O_2$ was changed to fresh DMEM with 20% FBS, and the cell culture was continued for 2 days. FAPs were then used for the following assay.

**Co-culture of RAW264.7 and FAPs**. RAW264.7 were labeled with CytoTrack™ Green (Bio-Rad), and Trp53(+/+) or Trp53(−/−) FAPs treated with or without $H_2O_2$ were labeled with CytoTrack™ Red (Bio-Rad). RAW264.7 ($8 \times 10^4$ cells/well) and Trp53(+/+) or Trp53(−/−) FAPs treated with or without $H_2O_2$ ($4 \times 10^4$ cells/well) were seeded on 12-well plates. The co-cultures were incubated for 72 h, and then cells were collected and analyzed on a FACSCanto™ II (BD Biosciences) equipped with 488- and 633-nm lasers. Data collection and analysis were conducted using FACSDiva™ (BD Biosciences) and Kaluza V1.5a (Beckman Coulter). RAW264.7 were purchased from American Type Culture Collection and tested for mycoplasma using the e-Myco™ Mycoplasma PCR Detection Kit (iNtRON Biotechnology Inc., Seongnam-si, South Korea).

**General experimental design and statistics**. Power calculations were not performed to predetermine the sample size. We did not use any specific method of randomization to determine how animals were allocated to experimental groups. Animals were excluded from the study only if their health status was compromised; for instance, if they had visible wounds due to fighting. The investigators were not blinded to allocations during experiments or outcome assessments, although they were blinded during the behavioral assessment.

Quantitative data are shown as means as well as medians with interquartile ranges (IQRs) and 1.5 times the IQR, and displayed by dot plots and box and whisker plots using ggplot2, which is a plotting system for R based on The Grammar of Graphics (The R Foundation for Statistical Computing, Vienna, Austria). Heat map, k-means clustering, and principal component analyses were performed by FactoMineR and factoextra, which are R packages. Normality was assessed using the Shapiro–Wilk test. The pairwise t test was used for comparison between two groups. A one-way analysis of variance (ANOVA) was conducted to assess differences among three groups or more. Pairwise comparisons were made only when the one-way ANOVA test indicated a statistical significance. P values for multiple comparisons were adjusted by the Holm method. Statistical analyses were performed using EZR, which is a graphical user interface for R[70]. Two-sided P values <0.05 were considered statistically significant.

**Reporting summary**. Further information on research design is available in the Nature Research Reporting Summary linked to this article.

## Data availability
The data that support the findings of this study are available from the corresponding author upon reasonable request.

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

## Acknowledgements

We would like to thank Dr. Mizue, Ms. Kamiya, Ms. Hayakawa, and Mr. Shiraishi for their technical support, as well as Zenis Co. Ltd. for providing native language editing services. We are also very grateful to Dr. Horio, Dr. Kuno, and Dr. Hosoda for providing the muscle tissues of mdx mice. This work supported by JSPS KAKENHI Grant Numbers JP16K16430 and JP18K17722, and LEOC Co. Ltd.

## Author contributions

Y.S. and T.S.C. designed all of the studies. Y.S. and T.S.C. conducted all experiments. Y.S. and T.M. performed the data analysis. M.N. bred and genotyped *B6.Cg-Trp53<tm1Sia>/Rbrc* mice. Y.S., T.S.C., and M.F. contributed to drafting and reviewing the manuscript.

## Competing interests

The authors declare no competing interests.
