## [Peer Review File · Nature Communications]

Reviewers' comments:

Reviewer #1 (Remarks to the Author):

This is an interesting manuscript that provides convincing evidence that (1) exercise induces a senescence profile in FAPs that promotes regeneration of muscle from injury; (2) that this senescence profile is impaired in a model of chronic inflammatory myopathy; and (3) that restoration of the exercise-induced senescence profile in FAPs can be achieved by supplementing the diet of mice with the AMPK activator AICAR. The findings are likely to be important to other investigators interested in muscle conditions associated with an impaired regenerative response to injury. Impact could be improved through inclusion of another model of impaired muscle regeneration, such as the mdx mouse (diaphragm muscle).

P. 12, 2nd para, lines 8-9: "but exercise in CIM mice...". The statement does not accurately reflect the results shown in Fig. 4 where exercise in CIM mice further reduced already low levels of senescent factors. Please rephrase accordingly.

P. 13, 1st para, last sentence: "Hence, we doubt whether pro-senescent intervention with FAPs will have as positive an effect as exercise therapy in chronic myopathy." This sentence does not seem to logically follow the preceding statement and also seems to contradict the experimental results showing that AICAR treatment (which promotes senescence of FAPs) in CIM restored the pro-senescence effect of exercise for FAPs and also induced significant benefits for muscle repair and whole body function. Please reconsider and revise accordingly.

Reviewer #2 (Remarks to the Author):

The manuscript by Saito et al explores the role of exercise in improving muscle regeneration in normal muscle and in myopathic muscle (mouse model of chronic inflammatory myopathy). The rationale for the study includes the observation that exercise can have both beneficial effects in inflammatory myopathies and detrimental effects (inflammation, fibrosis) in some myopathy patients and in normal controls. Because of the potential fibrotic effects, the authors focused on the effects of exercise on FAPs.

The authors use both an acute (AMI: BaCl₂ injury) and chronic (CMI: autoimmune myopathy) models of muscle damage, referring to the acute model as "regenerating" and the chronic model as "degenerating". However, neither of these models are pure degeneration or regeneration. The primary findings of the manuscript are that AMI results in FAP phenotypes that are pro-inflammatory and pro-apoptotic, where CMI results in FAP phenotypes that are anti-inflammatory and anti-apoptotic. For the exercise studies, the authors focus on the CMI model. Using a downhill running protocol, the authors provide evidence in support of the hypothesis that exercise increases senescence markers in FAPs from normal mice but not CIM mice. To test for the effects of promoting FAP senescence in the CMI mice, the authors administered AICAR

A primary conclusion of the study is that exercise increases the senescence phenotype of FAPs in normal mice but not in CIM mice. Another is that exercise plus AICAR improves the functional recovery in the CIM model by modulating the FAP phenotype to a more pro-apoptotic state. This would be an interesting and therapeutically relevant finding. However, I have several major concerns about the manuscript.

1. First and foremost, there is the problem of causality. There are no data in this manuscript to indicate that the differences in FAP phenotypes in response to different injuries, in response to exercise, or in response to AICAR are the cause of the different regenerative outcomes. These are strictly correlations, and it is certainly possible that the differences in regeneration in each case

are results of many interdependent processes involving myogenic cells, immune cells, etc.

2. In terms of the overall rationale for the study, on page 6 the authors state that "senescence is one of the primary regulators of cell apoptosis and clearance" and refer to reference #18. It is not clear what they mean by this. Senescence is neither a regulator of apoptosis, nor is a regulator of clearance.

3. In the absence of TNF-alpha ("control" in panel 2A), the authors provide no evidence that there is actually a difference in the endogenous FAP apoptosis in CIM vs AMI. The gene expression data are consistent with CMI FAPs exhibiting an "anti-apoptotic" phenotype, but there is no evidence that there is actually any less apoptosis of the FAPs in the CIM model.

4. In order to assess FAP senescence, the authors measure levels of p16 expression at the transcript level. While this may indeed be indicative of cellular senescence, elevation of p16 transcript alone is by no means sufficient evidence to label the FAPs "senescent". Likewise, in response to exercise, the authors rely on gene expression analysis to characterize the state of the cells.

5. In order to test the effects of exercise, the authors use forced downhill running, and paradigm designed to induce muscle damage and thus not relevant to any of the exercise paradigms that would be used in patients. If the goal is to test the effects of non-injurious physical activity (as would be the goal in humans), there are other protocols that would result in muscle exercise without damaging injuries.

Given these major concerns about the experimental design and data interpretation, it is difficult to draw any conclusions about effects of exercise (as occurs in humans) on muscle regeneration in response to acute or chronic injury, the role of FAPs as mediators of any exercise intervention, and whether AICAR is likely to be beneficial with or without exercise (except for injurious downhill running) in improving muscle regeneration

Other comments

1. In Fig. 1C, the authors stain for collagen and conclude that it is produced by FAPs. There is no data presented to support that conclusion.

2. In Fig. 2, the authors test for the sensitivity of FAPs in CMI and AMI to exogenous TNF-alpha. While indicative of susceptibility, these data do not demonstrate an actual increase in apoptosis in response to endogenous TNF-alpha in either model.

3. The authors claim that FAPs in the CIM model acquire a "clearance deficiency". This conclusion appears to be based on the persistence of FAPs in that model compared to the AMI model. However, is there any evidence to support that there is a clearance deficiency as opposed to a condition of persistent stimulation of activation and proliferation to account for the persistent number of FAPs? In any chronic model, there is ongoing damage that might be expected to continue to stimulate FAP proliferation, resulting in sustained cell numbers without any clearance deficiency.

Reviewer #3 (Remarks to the Author):

The paper by Saito and colleagues is a very interesting characterization of fibro-adipogenic progenitors (FAP) in normal and chronic inflammatory myopathy (CIM) models. The experiments characterizing FAPs in both models was quite thorough and informative. The overarching finding is that FAPs in normal mice acquire senescence and allow for appropriate regeneration and exercise adaptation. While FAPs in CIM are apoptosis resistant, do not turn on senescence and as a result regeneration is impaired as is exercise adaptation. These results are novel, and make significant

contributions to our understanding of muscle regeneration and the complex processes that govern regeneration. The study design and methods employed are appropriate and the results informative. Having said this, there are several key experiments that are missing that would ultimately identify the role that FAPs are playing skeletal muscle regeneration.

Major comments:

1. Although compelling there remains a gap in the connection between FAPs and their role in regeneration. If in fact there is a necessity for senescence of FAPs in the regenerative and adaptive processes of skeletal muscle then one would expect a reciprocal relationship between the FAP response and the muscle satellite cell response (i.e. lack of FAP senescence should result in impaired satellite cell function). There is a complete absence of the satellite cell response to regeneration in these experiments despite the fact that muscle satellite cell is the driving force behind regeneration. There needs to be some characterization of the FAP response coordinated to the satellite cell response to fully sell this story. In addition, although the authors discuss the importance of the inflammatory response for muscle regeneration there should be some characterization of the differences between models in the recruitment of immune and inflammatory cells in the post injury period. Together, analysis of the satellite cell population and the inflammatory cell population in conjunction with the FAP response would provide a much clearer picture of the importance of FAPs in the regenerative and adaptive process.

Minor comments:

2. Line 37-39 – The concerns are that excessive physical activity could result in significant muscle damage, which could result in downstream pathological events such as kidney failure in extreme cases. The concern is not that CPK will go up or that inflammatory cell infiltration will go up per se. These are simply surrogate markers of muscle damage.

3. It seems that the collagen analysis was simply qualitative as there is no quantitative analysis included in the manuscript. This should be quantitatively analyzed.

4. Is it sufficient to say that a decrease in the expression of senescent factors led to senescent cells? Is it possible that these cells were still active despite a decrease in the expression of senescent factors?

We thank the Reviewers for their careful consideration of our manuscript. We have revised the manuscript by taking into account each point raised by the reviewers. These changes are highlighted in “red” color fonts in the revised manuscript. We have now addressed each of the comments as outlined below.

Reviewers' comments:

Reviewer #1 (Remarks to the Author):

This is an interesting manuscript that provides convincing evidence that (1) exercise induces a senescence profile in FAPs that promotes regeneration of muscle from injury; (2) that this senescence profile is impaired in a model of chronic inflammatory myopathy; and (3) that restoration of the exercise-induced senescence profile in FAPs can be achieved by supplementing the diet of mice with the AMPK activator AICAR. The findings are likely to be important to other investigators interested in muscle conditions associated with an impaired regenerative response to injury. Impact could be improved through inclusion of another model of impaired muscle regeneration, such as the mdx mouse (diaphragm muscle).

Author Response: We would sincerely like to thank Reviewer #1 for the positive comments on our manuscript. We have now added immunohistochemistry data for PDGFR α and p16INK4A in AMI, CIM, and mdx muscle to test whether FAPs express p16INK4A or not. We found that the percentage of p16INK4A-positive FAPs in AMI mice was significantly higher than in CIM and mdx mice. Most of the FAPs in CIM and mdx mice did not express p16INK4A, but we could detect p16INK4A-positive/PDGFR α -negative cells in close contact with p16INK4A-negative FAPs (Fig. S4A-S4D). We have added these results in Figure S4A-S4D, and have described these points in the results section, lines 145-151, and in the discussion section, lines 381-383.

Next, we investigated the expression levels of β -klotho in FAPs isolated from AMI and CIM mice. Klotho is known to be an anti-aging protein and its expression level is negatively correlated with cell senescence; it has also been reported that klotho has an anti-inflammatory effect ^{1,2}. In mdx mice, it has been reported that β -klotho expression levels are upregulated and therefore β -klotho may be involved in muscle fibrosis ³, and β -klotho expression has been observed in FAPs but not satellite cells ⁴. In our results, β -klotho expression in CIM-FAPs was significantly up-regulated compared with AMI-FAPs. We next analyzed the correlation between β -klotho and p16INK4a in FAPs,

because β -klotho is a negative regulator of p16^{INK4A} and is strongly expressed by FAPs in skeletal muscle^{4,5}. In our study we found a strong negative correlation between β -klotho and p16INK4a. We have added this finding in Figure S4E and S4F, and this point has been detailed in the results section, lines 152-159, and in the discussion section, lines 381-383.

Demonstrating a further similarity between FAPs in CIM and mdx mice, Sohn et al. reported that in dystrophin/utrophin double-knockout mice, non-myogenic mesenchymal stem cells, which are PDGFR α -expressing interstitial cells, up-regulated wnt/ β -catenin signaling⁶. The wnt/ β -catenin activation promoted cyclin D1, which is a negative regulator of p16INK4A, and c-Myc, which is a positive regulator of PD-L1 and CD47^{7,8}. We confirmed the down-regulation of p16INK4A and the up-regulation of PD-L1 and CD47 in CIM-FAPs (please see the response to “Other Comment #3” by Reviewer #2), and we therefore hypothesize that FAPs in dystrophin/utrophin double-knockout mice may have an anti-apoptotic and low senescence phenotype, as in CIM-FAPs.

Normal FAPs promote satellite cell differentiation, but this effect is decreased in mdx-derived FAPs⁹. Therefore, to detect the similarity in FAP phenotypes between CIM and mdx mice, we conducted FAP–satellite cell transwell co-culture experiments and investigated whether FAPs derived from AMI or CIM mice differentially affected the ability to promote myogenesis in normal satellite cells. We found that FAPs isolated from AMI mice potentiated satellite cells to form MyHC-positive multinucleated myotubes, but FAPs isolated from CIM mice did not potentiate myogenesis (Fig. 3E-3H), a result that was previously demonstrated in mdx-FAPs⁹. We have added these results in Figure 3F-3H, and provided details in the results section, lines 141-144, and in the discussion section, lines 381-383.

P. 12, 2nd para, lines 8-9: “but exercise in CIM mice...”. The statement does not accurately reflect the results shown in Fig. 4 where exercise in CIM mice further reduced already low levels of senescent factors. Please rephrase accordingly.

Author Response: Thank you for your comment. We have rephrased the statement as follows: “but exercise in CIM mice decreased the expression of senescent factors to levels seen in sedentary CIM mice”. Please see the discussion section, lines 302-304.

P. 13, 1st para, last sentence: “Hence, we doubt whether pro-senescent intervention with FAPs will have as positive an effect as exercise therapy in chronic myopathy.” This sentence does not seem to logically follow the preceding statement and also seems to contradict the experimental results showing that AICAR treatment (which promotes senescence of FAPs) in CIM restored the pro-senescence effect of exercise for FAPs and also induced significant benefits for muscle repair and whole body function. Please reconsider and revise accordingly.

Author Response: Thank you for your suggestion, which we agree with. We have now revised the relevant material in the discussion (P. 13, 1st paragraph, last sentence) as indicated below.

“Hence, we tested whether the combined application of exercise and pro-senescent intervention targeting FAPs had a positive effect in mice with chronic myopathy.” Please see the discussion section, lines 315- 317.

Reviewer #2 (Remarks to the Author):

The manuscript by Saito et al explores the role of exercise in improving muscle regeneration in normal muscle and in myopathic muscle (mouse model of chronic inflammatory myopathy). The rationale for the study includes the observation that exercise can have both beneficial effects in inflammatory myopathies and detrimental effects (inflammation, fibrosis) in some myopathy patients and in normal controls. Because of the potential fibrotic effects, the authors focused on the effects of exercise on FAPs.

The authors use both an acute (AMI: BaCl₂ injury) and chronic (CMI: autoimmune myopathy) models of muscle damage, referring to the acute model as “regenerating” and the chronic model as “degenerating”. However, neither of these models are pure degeneration or regeneration. The primary findings of the manuscript are that AMI results in FAP phenotypes that are pro-inflammatory and pro-apoptotic, where CMI results in FAP phenotypes that are anti-inflammatory and anti-apoptotic. For the exercise studies, the authors focus on the CMI model. Using a downhill running protocol, the authors provide evidence in support of the hypothesis that exercise increases senescence markers in FAPs from normal mice but not CIM mice. To test for the effects of promoting FAP senescence in the CMI mice, the authors administered AICAR

A primary conclusion of the study is that exercises increases the senescence phenotype of FAPs in normal mice but not in CIM mice. Another is that exercise plus AICAR improves the functional recovery in the CIM model by modulating the FAP phenotype to a more pro-apoptotic state. This would be an interesting and therapeutically relevant finding. However, I have several major concerns about the manuscript.

Author Response: We are grateful to Reviewer #2 for the positive remarks concerning our manuscript. We have revised the text based on each comment, and we hope that our modifications have improved the manuscript.

1. First and foremost, there is the problem of causality. There are no data in this manuscript to indicate that the differences in FAP phenotypes in response to different injuries, in response to exercise, or in response to AICAR are the cause of the different regenerative outcomes. These are strictly correlations, and it is certainly possible that the differences in regeneration in each case are results of many interdependent processes involving myogenic cells, immune cells, etc.

Author response: Thank you for your comment. We demonstrated differences in FAP phenotypes in response to different injuries, exercise, and AICAR, and determined the subsequent effects on muscle regeneration by PCR array and PCA analysis; these results are shown in Fig. 6E and 6F. Further, the results of mouse behavioral assessments are shown in Fig. 5B-5D. As you pointed out, additional data are required to prove the presence of a causal relation between FAP senescence and regulation of muscle regeneration. However, in our in vitro experiments, CIM-FAPs that expressed low levels of p16INK4A and γ H2AX (Fig. 3A and 3C, and Fig. S3) also expressed low levels of inflammatory cytokines (Fig. 2D and 2E) and high levels of PD-L1/2 and CD47 (which facilitate escape from immunoclearance) (Fig. 2G), and impaired the promotion of satellite cell differentiation (Fig. 3E-3H). On the other hand, AMI-FAPs that expressed high levels of p16INK4A and γ H2AX (Fig. 3A and 3C, and Fig. S3) expressed high levels of inflammatory cytokines (Fig. 2D and 2E) and low levels of PD-L1/2 and CD47 (Fig. 2G), and promoted satellite cell differentiation (Fig. 3E-3H). Previous studies showed that senescence is causally essential for the suppression and regeneration of tissue fibrosis by demonstrating that senescent fibroblasts or hepatic stellate cells inhibited tissue fibrosis and promoted tissue regeneration, and that knockdown or knockout of p16INK4a or p53 in

these cells promoted tissue fibrosis and inhibited tissue regeneration¹⁰⁻¹². Like FAPs, fibroblasts and hepatic stellate cells are mesenchymal in origin. Our results showed that p16INK4A in FAPs was highly expressed in AMI mice, an animal model in which there is sufficient muscle regeneration and no fibrosis, but p16INK4A showed lower expression in CIM mice, a model characterized by fibrosis and muscle degeneration. We showed a correlation between senescence of FAPs and muscle regeneration, and this finding, in combination with the results of the previous studies cited above¹⁰⁻¹², indicate that senescent FAPs may regulate muscle regeneration during inflammation. It is possible that the senescence of FAPs is important in muscle regeneration due to the strong influence of FAP senescence on cytokine expression, cell clearance, apoptosis ability, and promotion of satellite cell differentiation. We raised these issues in the discussion section, line 356-377.

2. In terms of the overall rationale for the study, on page 6 the authors state that “senescence is one of the primary regulators of cell apoptosis and clearance” and refer to reference #18. It is not clear what they mean by this. Senescence is neither a regulator of apoptosis, nor is a regulator of clearance.

Author response: Thank you for your comment. We have modified the following text in the discussion section, lines 232-252, as follows: “In response to a variety of stresses, mammalian cells undergo senescence. Although the crucial determinants of whether a cell responds to damage by undergoing cell survival or cell death (apoptosis) are the cell type and the nature and intensity of the damage, apoptosis is sometimes a response to acute/high stress, whereas cell survival can result from low/constitutive stress in cells with low levels of senescent factors¹³. For example, p53, a senescence cell marker, is known to trigger apoptosis in response to cellular stress, but whether or not this occurs depends on stress and cell type, specific modulated genes, p53 levels, and transcriptional activity¹⁴. Low levels of p53 expression protect fibroblasts from the induction of apoptosis, and wild-type p53 expressed by embryonic fibroblasts induces apoptosis caused by ultraviolet-B-induced damage¹⁵; however, expression of the hypomorphic R172P p53 mutation abrogates p53-mediated apoptosis both by downregulating the pro-apoptotic factors PUMA and NOXA and inducing high expression levels of the pro-survival gene BCL-2 while keeping cell cycle control mostly intact, and enhances inflammation and

immunosuppression relative to wild-type mice¹⁶. Similarly, a recent study showed that downregulation of Bcl-2 was associated with p16-mediated apoptosis in non-small-cell lung cancer cells¹⁷. Another study demonstrated that p16 reactivation downregulated the expression of survivin, a crucial apoptotic regulator, and exhibited antitumor potency by downregulating AKt/survivin signaling in hepatocellular carcinoma cells¹⁸. In our study, expression levels of wild-type p53 and p16 were increased in AMI-FAPs compared with CIM-FAPs, with concomitant upregulation of BCL-2.”

During remodeling in stressed or injured tissues, senescent cells are detectable by neighboring cells, which promotes their elimination and rapid replacement by the homeostatic protective system. In our study, we also demonstrated that expression of PDL-1/2 and CD47 was upregulated in CIM-FAPs compared with AIM-FAPs. PDL-1/2 binds to PD-1 on activated immune cells to inhibit these cells’ activation and effector response, thus serving as a critical “don’t find me signal” to the adaptive immune system. In contrast, CD47 provides a “don’t eat me signal” to avoid phagocytosis by immune cells^{19,20}. The anti-apoptotic nature of the accumulated FAPs in CIM mice might impair the elimination and replacement of senescent cells. We have revised the discussion section, lines 279-292, to convey these points.

3. In the absence of TNF-alpha (“control” in panel 2A), the authors provide no evidence that there is actually a difference in the endogenous FAP apoptosis in CIM vs AMI. The gene expression data are consistent with CMI FAPs exhibiting an “anti-apoptotic” phenotype, but there is no evidence that there is actually any less apoptosis of the FAPs in the CIM model.

Author Response: Thank you for your comment. Although in the absence of TNF-alpha there was no difference between CIM-FAPs and AIM-FAPs in the expression of apoptotic markers (LDH and annexin; Figure 2A and 2B), we not only showed that CIM-FAPs had an anti-apoptotic phenotype, but also demonstrated expression of TNF receptor superfamilies, such as Fas, CD27, Tnfrsf14, and Tnfrsf19, which were increased in AIM-FAPs and decreased in CIM-FAPs in Figure 2D and 2E. The TNF signaling pathway mediates cell apoptosis, as indicated in the revised discussion, lines 263-265. These results suggest that TNF-alpha stimulation might affect the apoptosis of AIM-FAPs and CIM-FAPs. Therefore, we stimulated FAPs with TNF-alpha in both AIM and CMI mice, and demonstrated a difference in apoptosis rates between AIM-FAPs and CMI-FAPs in

vitro. Other studies also showed that cell apoptosis occurred following stimulation by TNF-alpha but not in its absence^{21,22}.

4. In order to assess FAP senescence, the authors measure levels of p16 expression at the transcript level. While this may indeed be indicative of cellular senescence, elevation of p16 transcript alone is by no means sufficient evidence to label the FAPs “senescent”. Likewise, in response to exercise, the authors rely on gene expression analysis to characterize the state of the cells.

Author Response: Thank you for your comment. We have now measured γ H2AX expression in both CIM-FAPs and AIM-FAPs by flow cytometry analysis (Fig. S3A). We found that H2A.X (γ H2A.X) expression was increased in AIM-FAPs compared with CIM-FAPs (Fig. S3B). γ H2A.X, which results from phosphorylation of the Ser-139 residue of the histone variant H2A.X, is an early cellular response to the induction of DNA double-strand breaks²³. This phosphorylation event is one of the most established chromatin modifications linked to DNA damage and repair²⁴. Activation of a DNA damage response, including formation of DNA damage foci containing activated H2A.X at either uncapped telomeres or persistent DNA double-strand breaks, is known to be the major trigger of cell senescence²⁵, and therefore γ H2A.X is used as a reliable quantitative indicator of senescent cells²⁶. We have added the above results in Figure S3A and S3B and have presented them in the results section, lines 124-132.

5. In order to test the effects of exercise, the authors use forced downhill running, and paradigm designed to induce muscle damage and thus not relevant to any of the exercise paradigms that would be used in patients. If the goal is to test the effects of non-injurious physical activity (as would be the goal in humans), there are other protocols that would result in muscle exercise without damaging injuries.

Author Response: Thank you for your comment. Low-intensity exercise for chronic myopathy has been preferred because of the fear that exercise will aggravate muscle inflammation, but a human study recently demonstrated that intensive exercise had positive effects in chronic myopathy²⁷. Transient inflammation after intensive exercise is one of the triggers of muscle regeneration²⁸, and therefore intensive exercise for chronic myopathy has attracted attention²⁷. Inflammation is required to change the intramuscular environment in chronic myopathy from a degenerative to a regenerative state²⁹, and

non-injurious physical exercise can limit the effect of exercise to only a minor improvement in motor function²⁷. Hence, when investigating new myopathy treatments in humans, it will be beneficial to assess the effects of exercise associated with muscle damage rather than only non-injurious physical exercise. However, the detailed mechanisms of the regeneration or degeneration/fibrosis induced by intensive exercise remain unknown. Therefore, to clarify this issue we used downhill running to induce muscle damage, with the hope that this research will lead to new myopathy treatments that are both safe and effective.

Given these major concerns about the experimental design and data interpretation, it is difficult to draw any conclusions about effects of exercise (as occurs in humans) on muscle regeneration in response to acute or chronic injury, the role of FAPs as mediators of any exercise intervention, and whether AICAR is likely to be beneficial with or without exercise (except for injurious downhill running) in improving muscle regeneration

Other comments

1. In Fig. 1C, the authors stain for collagen and conclude that it is produced by FAPs. There is no data presented to support that conclusion.

Author Response: Thank you for your comment. We have deleted the conclusion regarding collagen production by FAPs because there were no supporting data. Please see the results section, lines 79-81.

2. In Fig. 2, the authors test for the sensitivity of FAPs in CMI and AMI to exogenous TNF-alpha. While indicative of susceptibility, these data do not demonstrate an actual increase in apoptosis in response to endogenous TNF-alpha in either model.

Author Response: Thank you for your comment. A previous study by Lemos et al. examined the effect of TNF-alpha on FAP apoptosis in vivo^{30,31}. TNF-alpha-expressing macrophages in acute muscle inflammation induced FAP apoptosis, whereas in chronic inflammation, TGF-beta-expressing macrophages resulted in FAP accumulation, suggesting that FAP apoptosis might be positively correlated with the concentration of

TNF-alpha. To investigate this question, we tested FAPs in both AMI and CIM mice with varying doses of TNF-alpha in vitro. In the AIM model, the percentage of FAPs that underwent apoptosis was positively correlated with the TNF-alpha dose, but in the CIM model, the percentage of apoptotic FAPs did not vary with this dose. This suggests that in CIM mice, FAPs might maintain an anti-apoptotic phenotype regardless of the concentration of endogenous TNF-alpha. Regarding the response of FAPs to endogenous TNF-alpha, it was reported that FAP apoptosis occurred in AMI in the presence of TNF-alpha, but did not occur in vivo when TNF-alpha was inhibited by anti-TNF-alpha antibody; this phenomenon also occurred in vitro³⁰. These results were similar to our own in vitro findings. Therefore, our in vitro experiments that demonstrated different phenotypes of FAPs from AIM and CIM mice should not differ from the actual response in vivo.

3. The authors claim that FAPs in the CIM model acquire a “clearance deficiency”. This conclusion appears to be based on the persistence of FAPs in that model compared to the AMI model. However, is there any evidence to support that there is a clearance deficiency as opposed to a condition of persistent stimulation of activation and proliferation to account for the persistent number of FAPs? In any chronic model, there is ongoing damage that might be expected to continue to stimulate FAP proliferation, resulting in sustained cell numbers without any clearance deficiency.

Author Response: Thank you for your comment. We agree that the statement about clearance deficiency was based on the persistence of FAPs in that model compared to the AMI model. To provide additional evidence to support the conclusion that there is a clearance deficiency in CIM-FAPs, we have added new data on mRNA expression of programmed death ligand 1(PD-L1), PD-L2, and CD47 in FAPs. The clearance deficiency of diseased cells, e.g., tumor cells and vascular smooth muscle cells in atherosclerosis, is thought to be caused by impairment of immune surveillance machinery^{8,20,32}. We found that the mRNA expression of PD-L1/2 and CD47 was significantly up-regulated in CIM-FAPs compared to AMI-FAPs (Fig. 2G). These results support our hypothesis that there is a clearance deficiency in CIM-FAPs, because PD-L1/2 and CD47 produce anti-phagocytic signals, referred to as “don’t find me” and “don’t eat me” signals, respectively⁸. Therefore, the CIM-derived FAPs with PD-L1/2 and CD47 easily accumulated as a result of escaping from immunoclearance, as well as due to proliferation resulting from ongoing damage in chronic inflammation. We have added this information in Figure 2G, and this point has been made in the results section, lines 105-113, and the

discussion section, lines 279-292.

Reviewer #3 (Remarks to the Author):

The paper by Saito and colleagues is a very interesting characterization of fibro-adipogenic progenitors (FAP) in normal and chronic inflammatory myopathy (CIM) models. The experiments characterizing FAPs in both models was quite thorough and informative. The overarching finding is that FAPs in normal mice acquire senescence and allow for appropriate regeneration and exercise adaptation. While FAPs in CIM are apoptosis resistant, do not turn on senescence and as a result regeneration is impaired as is exercise adaptation. These results are novel, and make significant contributions to our understanding of muscle regeneration and the complex processes that govern regeneration. The study design and methods employed are appropriate and the results informative. Having said this, there are several key experiments that are missing that would ultimately identify the role that FAPs are playing skeletal muscle regeneration.

Author Response: We sincerely thank the Reviewer #3 for the encouraging comments on our manuscript. We have addressed all the comments/points raised by Reviewer #3 and believe these additions have improved the quality of the manuscript.

Major comments:

1. Although compelling there remains a gap in the connection between FAPs and their role in regeneration. If in fact there is a necessity for senescence of FAPs in the regenerative and adaptive processes of skeletal muscle then one would expect a reciprocal relationship between the FAP response and the muscle satellite cell response (i.e. lack of FAP senescence should result in impaired satellite cell function). There is a complete absence of the satellite cell response to regeneration in these experiments despite the fact that muscle satellite cell is the driving force behind regeneration. There needs to be some characterization of the FAP response coordinated to the satellite cell response to fully sell this story. In addition, although the authors discuss the importance of the inflammatory response for muscle regeneration there should be some characterization of the differences between models in the recruitment of immune and inflammatory cells in the post injury period. Together, analysis of the satellite cell population and the inflammatory cell population in

conjunction with the FAP response would provide a much clearer picture of the importance of FAPs in the regenerative and adaptive process.

Author Response: Thank you for this important consideration. We agree that satellite cells and inflammatory cells play an important role in the regenerative process. We performed FAP-satellite cell transwell co-culture experiments to test whether the difference in FAPs derived from AMI vs. CIM mice affects the ability of these cells to promote myogenesis in normal satellite cells. We found that FAPs isolated from AMI but not CIM mice potentiated satellite cells to form MyHC-positive multinucleated myotubes (Fig. 3F-3H). We have added these results in Figure 3F-3H, and have detailed these findings in the results section, lines 141-144. Further, senescent FAPs derived from AMI mice or exercised muscle demonstrated a higher expression of IL-33 and TSG-6 compared with FAPs derived from CIM mice, and IL-33 and TSG-6 promote muscle regeneration through regulatory T cell, macrophage, and myoblasts³³⁻³⁶, as described in the discussion section. We have also presented new results about the CD11b-positive cell population, which includes monocytes, macrophages, NK cells, and granulocytes, in AMI and CIM mice. The percentage of CD11b+ cells in CIM mice was lower compared with that in AMI mice (Fig. S2A and S2B). We had already indicated that the expression of p16^{INK4A} and p53 and the senescence-associated secretory phenotype (SASP) was lower in CIM-FAPs than that in AMI-FAPs (Fig. 2D and 2E, and Fig. 3B and 3C), and it is well-known that one of the major functions of the SASP is to recruit immune cells to eliminate the senescent cells³⁷. The inhibition of CD11b+ cell recruitment in CIM mice might be caused by weak SASP expression in CIM-FAPs, which might result in insufficient regeneration in CIM mice³⁸. This has been detailed in the results section, lines 101-103, and the discussion section, lines 259-262. In addition, we have added new results about immune checkpoints (please see the response to “Other Comments #3 by Reviewer #2) in Figure 2G, and described them in the results section, lines 105-113, and in the discussion section, lines 279-292. We showed that PD-L1/2 and CD47 were upregulated in CIM-FAPs compared with AMI-FAPs. The upregulation of these factors was particularly pronounced in tumor cells and myofibroblasts^{20,39}, and they contributed to cell accumulation both by facilitating escape from the immune system and by inducing anti-phagocytic signaling, referred to as “don’t find me” and “don’t eat me” signals, respectively⁸. In conclusion, we discussed the idea that AMI-FAPs could recruit many immune cells and could also be cleared by phagocytic cells; in contrast, CIM-FAPs could easily accumulate as a result of weak recruitment of immune cells and escape from the immune system, due to downregulation

of the expression levels of p16^{INK4A} and p53 and upregulation of the expression levels of PD-L1/2 and CD47.

Minor comments:

2. *Line 37-39 – The concerns are that excessive physical activity could result in significant muscle damage, which could result in downstream pathological events such as kidney failure in extreme cases. The concern is not that CPK will go up or that inflammatory cell infiltration will go up per se. These are simply surrogate markers of muscle damage.*

Author Response: Thank you for your comment, which we agree with. We have changed the text to indicate that excessive physical exercise could result in significant muscle damage, potentially causing acute renal failure, liver dysfunction, compartment syndrome, heart failure, arrhythmias, electrolyte imbalance, and in severe cases, death⁴⁰. The new material can be found in the revised manuscript on lines 38-40 in the introduction section.

3. *It seems that the collagen analysis was simply qualitative as there is no quantitative analysis included in the manuscript. This should be quantitatively analyzed.*

Author Response: Thank you for your comment. We have now quantified collagen analysis (Fig. 1E), and this has been detailed in the results, lines 79-81.

4. *Is it sufficient to say that a decrease in the expression of senescent factors led to senescent cells? Is it possible that these cells were still active despite a decrease in the expression of senescent factors?*

Author Response: Thank you for your comment. Regarding senescent factors and senescent cells, please see our response to “Major Comment #4” by Reviewer #2. Cells expressing senescent factors, such as P16^{INK4A} and p53, may develop the SASP, causing them to secrete hundreds of factors that promote immunoclearance, including proinflammatory cytokines, chemokines, growth factors, and proteases³⁷. If the cells expressing senescent factors are defined as “active,” AIM-FAPs are active, but not

CIM-FAPs, because the CIM-FAPs show decreased expression of senescent factors and development of the SASP, as shown in Figure 2D and 2E. Recent studies demonstrated that senescent factor-expressing cells accumulate with aging and in chronic fibrotic disease. Although these senescent cells express senescent factors such as p16 and p53, fewer cells develop the SASP. In our study, aging was not an investigated factor, but if this were added to our model, the senescent cell phenotype might be different.

References

1. Kurosu, H. *et al.* Suppression of aging in mice by the hormone Klotho. *Science* **309**, 1829–1833 (2005).
2. Liu, F., Wu, S., Ren, H. & Gu, J. Klotho suppresses RIG-I-mediated senescence-associated inflammation. *Nat. Cell Biol.* **13**, 254–262 (2011).
3. Stuelsatz, P., Keire, P., Almuly, R. & Yablonka-Reuveni, Z. A contemporary atlas of the mouse diaphragm: myogenicity, vascularity, and the Pax3 connection. *J. Histochem. Cytochem.* **60**, 638–657 (2012).
4. Phelps, M., Stuelsatz, P. & Yablonka-Reuveni, Z. Expression profile and overexpression outcome indicate a role for β Klotho in skeletal muscle fibro/adipogenesis. *FEBS J.* **283**, 1653–1668 (2016).
5. Sato, S. *et al.* Ablation of the p16(INK4a) tumour suppressor reverses ageing phenotypes of klotho mice. *Nat Commun* **6**, 7035 (2015).
6. Sohn, J., Lu, A., Tang, Y., Wang, B. & Huard, J. Activation of non-myogenic mesenchymal stem cells during the disease progression in dystrophic dystrophin/utrophin knockout mice. *Hum. Mol. Genet.* **24**, 3814–3829 (2015).
7. Gong, A. & Huang, S. FoxM1 and Wnt/ β -catenin signaling in glioma stem cells. *Cancer Res.* **72**, 5658–5662 (2012).
8. Casey, S. C. *et al.* MYC regulates the antitumor immune response through CD47 and PD-L1. *Science* **352**, 227–231 (2016).
9. Mozzetta, C. *et al.* Fibroadipogenic progenitors mediate the ability of HDAC inhibitors to promote regeneration in dystrophic muscles of young, but not old Mdx mice. *EMBO Mol Med* **5**, 626–639 (2013).
10. Jun, J.-I. & Lau, L. F. The matricellular protein CCN1 induces fibroblast senescence and restricts fibrosis in cutaneous wound healing. *Nat. Cell Biol.* **12**, 676–685 (2010).

11. Krizhanovsky, V. *et al.* Senescence of activated stellate cells limits liver fibrosis. *Cell* **134**, 657–667 (2008).
12. Demaria, M. *et al.* An essential role for senescent cells in optimal wound healing through secretion of PDGF-AA. *Dev. Cell* **31**, 722–733 (2014).
13. Childs, B. G., Baker, D. J., Kirkland, J. L., Campisi, J. & van Deursen, J. M. Senescence and apoptosis: dueling or complementary cell fates? *EMBO Rep.* **15**, 1139–1153 (2014).
14. Chen, Q. M., Liu, J. & Merrett, J. B. Apoptosis or senescence-like growth arrest: influence of cell-cycle position, p53, p21 and bax in H₂O₂ response of normal human fibroblasts. *Biochem. J.* **347**, 543–551 (2000).
15. Lassus, P., Ferlin, M., Piette, J. & Hibner, U. Anti-apoptotic activity of low levels of wild-type p53. *EMBO J.* **15**, 4566–4573 (1996).
16. Tavana, O. *et al.* Absence of p53-dependent apoptosis leads to UV radiation hypersensitivity, enhanced immunosuppression and cellular senescence. *Cell Cycle* **9**, 3328–3336 (2010).
17. Kataoka, M. *et al.* Down-regulation of bcl-2 is associated with p16INK4-mediated apoptosis in non-small cell lung cancer cells. *Oncogene* **19**, 1589–1595 (2000).
18. Hu, H. *et al.* P16 reactivation induces anoikis and exhibits antitumour potency by downregulating Akt/survivin signalling in hepatocellular carcinoma cells. *Gut* **60**, 710–721 (2011).
19. Chao, M. P., Weissman, I. L. & Majeti, R. The CD47-SIRP α pathway in cancer immune evasion and potential therapeutic implications. *Curr. Opin. Immunol.* **24**, 225–232 (2012).
20. Zou, W., Wolchok, J. D. & Chen, L. PD-L1 (B7-H1) and PD-1 pathway blockade for cancer therapy: Mechanisms, response biomarkers, and combinations. *Sci Transl Med* **8**, 328rv4–328rv4 (2016).
21. Maratheftis, C. I., Andreakos, E., Moutsopoulos, H. M. & Voulgarelis, M. Toll-like Receptor-4 Is Up-Regulated in Hematopoietic Progenitor Cells and Contributes to Increased Apoptosis in Myelodysplastic Syndromes. *Clinical Cancer Research* **13**, 1154–1160 (2007).
22. Benderska, N. *et al.* DAPK-HSF1 interaction as a positive-feedback mechanism stimulating TNF-induced apoptosis in colorectal cancer cells. *J. Cell. Sci.* **127**, 5273–5287 (2014).
23. Mah, L.-J., El-Osta, A. & Karagiannis, T. C. gammaH2AX: a sensitive molecular marker of DNA damage and repair. *Leukemia* **24**, 679–686 (2010).
24. Sedelnikova, O. A., Rogakou, E. P., Panyutin, I. G. & Bonner, W. M. Quantitative detection of (125)IdU-induced DNA double-strand breaks with gamma-H2AX antibody. *Radiat. Res.* **158**, 486–492 (2002).
25. d'Adda di Fagagna, F. *et al.* A DNA damage checkpoint response in telomere-initiated senescence. *Nature* **426**, 194–198 (2003).

26. Wang, C. *et al.* DNA damage response and cellular senescence in tissues of aging mice. *Aging Cell* **8**, 311–323 (2009).
27. Alexanderson, H. Physical exercise as a treatment for adult and juvenile myositis. *J. Intern. Med.* **280**, 75–96 (2016).
28. Sciorati, C., Rigamonti, E., Manfredi, A. A. & Rovere-Querini, P. Cell death, clearance and immunity in the skeletal muscle. *Cell Death Differ.* **23**, 927–937 (2016).
29. Eming, S. A., Wynn, T. A. & Martin, P. Inflammation and metabolism in tissue repair and regeneration. *Science* **356**, 1026–1030 (2017).
30. Lemos, D. R. *et al.* Nilotinib reduces muscle fibrosis in chronic muscle injury by promoting TNF-mediated apoptosis of fibro/adipogenic progenitors. *Nat. Med.* **21**, 786–794 (2015).
31. Tidball, J. G. & Wehling-Henricks, M. Shifts in macrophage cytokine production drive muscle fibrosis. *Nat. Med.* **21**, 665–666 (2015).
32. Kojima, Y. *et al.* CD47-blocking antibodies restore phagocytosis and prevent atherosclerosis. *Nature* **536**, 86–90 (2016).
33. Kuswanto, W., Burzyn, D., Panduro, M., Wang, K. K. & Jang, Y. C. Poor Repair of Skeletal Muscle in Aging Mice Reflects a Defect in Local, Interleukin-33-Dependent Accumulation of Regulatory T Cells. *Immunity* **44**, 355–367 (2016).
34. Burzyn, D. *et al.* A special population of regulatory T cells potentiates muscle repair. *Cell* **155**, 1282–1295 (2013).
35. Castiglioni, A. *et al.* FOXP3+ T Cells Recruited to Sites of Sterile Skeletal Muscle Injury Regulate the Fate of Satellite Cells and Guide Effective Tissue Regeneration. *PLoS ONE* **10**, e0128094 (2015).
36. Fujita, R. *et al.* Endogenous mesenchymal stromal cells in bone marrow are required to preserve muscle function in mdx mice. *Stem Cells* **33**, 962–975 (2015).
37. McHugh, D. & Gil, J. Senescence and aging: Causes, consequences, and therapeutic avenues. *J. Cell Biol.* **217**, 65–77 (2018).
38. Arnold, L. *et al.* Inflammatory monocytes recruited after skeletal muscle injury switch into antiinflammatory macrophages to support myogenesis. *J. Exp. Med.* **204**, 1057–1069 (2007).
39. Gordon, S. R. *et al.* PD-1 expression by tumour-associated macrophages inhibits phagocytosis and tumour immunity. *Nature* **545**, 495–499 (2017).
40. Kim, J. *et al.* Exercise-induced rhabdomyolysis mechanisms and prevention: A literature review. *Journal of Sport and Health Science* **5**, 324–333 (2015).

Reviewers' comments:

Reviewer #1 (Remarks to the Author):

The authors have satisfactorily addressed my prior concerns.

Reviewer #3 (Remarks to the Author):

Thank you for carefully considering my comments and thoroughly addressing them in the manuscript.

We have revised the manuscript by taking into account each point raised by Reviewer #2. These changes are indicated by blue text in this document and in the revised manuscript, and previously changed contents are indicated by red text in the revised manuscript. We addressed each of the comments as outlined below.

Reviewers' comments:

Reviewer #2 (Remarks to the Author):

The manuscript by Saito et al explores the role of exercise in improving muscle regeneration in normal muscle and in myopathic muscle (mouse model of chronic inflammatory myopathy). The rationale for the study includes the observation that exercise can have both beneficial effects in inflammatory myopathies and detrimental effects (inflammation, fibrosis) in some myopathy patients and in normal controls. Because of the potential fibrotic effects, the authors focused on the effects of exercise on FAPs.

The authors use both an acute (AMI: BaCl₂ injury) and chronic (CMI: autoimmune myopathy) models of muscle damage, referring to the acute model as “regenerating” and the chronic model as “degenerating”. However, neither of these models are pure degeneration or regeneration. The primary findings of the manuscript are that AMI results in FAP phenotypes that are pro-inflammatory and pro-apoptotic, where CMI results in FAP phenotypes that are anti-inflammatory and anti-apoptotic. For the exercise studies, the authors focus on the CMI model. Using a downhill running protocol, the authors provide evidence in support of the hypothesis that exercise increases senescence markers in FAPs from normal mice but not CIM mice. To test for the effects of promoting FAP senescence in the CMI mice, the authors administered AICAR

A primary conclusion of the study is that exercises increases the senescence phenotype of FAPs in normal mice but not in CIM mice. Another is that exercise plus AICAR improves the functional recovery in the CIM model by modulating the FAP phenotype to a more pro-apoptotic state. This would be an interesting and therapeutically relevant finding. However, I have several major concerns about the manuscript.

Author Response: We are grateful for the comments of Reviewer #2. We have performed additional experiments using Trp53 knockout mice to indicate that FAP senescence is causally necessary to promote muscle regeneration. We hope that our modifications have improved the manuscript.

1. First and foremost, there is the problem of causality. There are no data in this manuscript to indicate that the differences in FAP phenotypes in response to different injuries, in response to exercise, or in response to AICAR are the cause of the different regenerative outcomes. These are strictly correlations, and it is certainly possible that the differences in regeneration in each case are results of many interdependent processes involving myogenic cells, immune cells, etc.

Author Response: We have added additional experiments using Trp53 knockout mice to demonstrate that FAP senescence is causally necessary for promoting muscle regeneration. First, we isolated FAPs from Trp53(+/+) mice and Trp53(-/-) mice, and these cells were transplanted into the muscles of normal mice (Fig. 4A). One week after transplantation, BaCl₂ solution was injected into the triceps surae to induce acute muscle injury. Regeneration of muscles transplanted with Trp53(+/+) FAPs occurred 20 days after injury; in contrast, muscles transplanted with Trp53(-/-) FAPs showed accumulation of FAPs and inflammatory cells at 10 days after injury, and muscle interstitial fibrosis at 20 days after injury (Fig. 4B-G). To better understand the mechanism whereby Trp53(-/-) FAPs mediated the impairment of muscle regeneration, we performed an in vitro study using a model involving H₂O₂ stimulation, because the production of reactive oxygen species in skeletal muscle has been shown to occur in muscle injury, including that induced by exercise^{1,2}. H₂O₂-treated Trp53(+/+) FAPs induced P21 and P16INK4A expression and showed a muscle regenerative phenotype with upregulated follistatin expression; these cells also promoted the C2C12 differentiation. On the other hand, Trp53(-/-) FAPs did not upregulate the expression of p21 and P16INK4A, and showed an immune escape phenotype, which was confirmed by PD-L1 and CD47 expression and a phagocytosis assay. Further, H₂O₂-treated Trp53(-/-) FAPs did not promote C2C12 differentiation. Therefore, damage-induced FAP senescence is necessary to promote muscle regeneration. These additional results are shown in Figure 4 and in the results section on pages 11-13, lines 178-215.

2. In terms of the overall rationale for the study, on page 6 the authors state that “senescence is one of the primary regulators of cell apoptosis and clearance” and refer to reference #18. It is not clear what they mean by this. Senescence is neither a regulator of apoptosis, nor is a regulator of clearance.

Author Response: Thank you for your comment. We have modified relevant text in the discussion section, lines 289-317, as follows: “In response to a variety of stresses, mammalian cells undergo senescence. Although the crucial determinants of whether a cell responds to damage by undergoing cell survival or cell death (apoptosis) are the cell type and the nature and intensity of the damage, apoptosis is sometimes a response to acute/high stress, whereas cell survival can result from low/constitutive stress in cells with low levels of senescent factors³. For example, p53, a senescence cell marker, is known to trigger apoptosis in response to cellular stress, but whether or not this occurs depends on stress and cell type, specific modulated genes, p53 levels, and transcriptional activity⁴. Low levels of p53 expression protect fibroblasts from the induction of apoptosis, and wild-type p53 expressed by embryonic fibroblasts induces apoptosis caused by ultraviolet-B-induced damage⁵; however, expression of the hypomorphic R172P p53 mutation abrogates p53-mediated apoptosis both by downregulating the pro-apoptotic factors PUMA and NOXA and inducing high expression levels of the pro-survival gene Bcl-2 while keeping cell cycle control mostly intact, and enhances inflammation and immunosuppression relative to wild-type mice⁶. Similarly, a recent study showed that downregulation of Bcl-2 was associated with p16-mediated apoptosis in non-small-cell lung cancer cells⁷. Another study demonstrated that p16 reactivation downregulated the expression of survivin, a crucial apoptotic regulator, and exhibited antitumor potency by downregulating AKt/survivin signaling in hepatocellular carcinoma cells⁸. In our study, expression levels of p53 and p16 were increased in AMI-FAPs compared with CIM-FAPs, with concomitant upregulation of Bcl-2.”

Further, Trp53(-/-) FAPs showed an anti-apoptotic effect in response to H₂O₂ or TNF- α stimulation in vitro. During remodeling in stressed or injured tissues, senescent cells are detectable by neighboring cells, which promotes their elimination and rapid replacement by the homeostatic protective system. In our study, we also demonstrated that the expressions of PDL-1/2 and CD47 were upregulated in CIM-FAPs compared with AIM-FAPs. PDL-1/2 binds to PD-1 on activated immune cells to inhibit these cells' activation and effector responses, thus serving as a critical “don't find me signal” for the adaptive immune system. In contrast, CD47 provides a “don't eat me signal” to avoid phagocytosis by immune cells^{9,10}. The anti-apoptotic nature of the accumulated FAPs in CIM mice might impair the elimination and replacement of senescent cells. We have revised the discussion section, lines 289-317, to convey these points.

3. *In the absence of TNF-alpha (“control” in panel 2A), the authors provide no evidence that there is actually a difference in the endogenous FAP apoptosis in CIM vs AMI. The gene expression data are consistent with CMI FAPs exhibiting an “anti-apoptotic” phenotype, but there is no evidence that there is actually any less apoptosis of the FAPs in the CIM model.*

Author Response: We performed active caspase-3 staining, which provided evidence demonstrating a difference in endogenous FAP apoptosis in CIM and AMI. We found that apoptotic FAPs positive for active caspase-3 were significantly increased in the AMI model but not the CIM model. Please see Figure 2F and 2G, and the results section, page 7, lines 109-110.

4. *In order to assess FAP senescence, the authors measure levels of p16 expression at the transcript level. While this may indeed be indicative of cellular senescence, elevation of p16 transcript alone is by no means sufficient evidence to label the FAPs “senescent”. Likewise, in response to exercise, the authors rely on gene expression analysis to characterize the state of the cells.*

Author Response: Thank you for your comment. We have now measured γ H2AX expression in both CIM-FAPs and AIM-FAPs by flow cytometry analysis (Fig. 3D and 3E). We found that H2A.X (γ H2A.X) expression was increased in AIM-FAPs compared with CIM-FAPs (Fig. 3D and 3E). γ H2A.X, which results from phosphorylation of the Ser-139 residue of the histone variant H2A.X, is an early cellular response to the induction of DNA double-strand breaks¹¹. This phosphorylation event is one of the most established chromatin modifications linked to DNA damage and repair¹². Activation of a DNA damage response, including formation of DNA damage foci containing activated H2A.X at either uncapped telomeres or persistent DNA double-strand breaks, is known to be the major trigger of cell senescence¹³, and therefore γ H2A.X is used as a reliable quantitative indicator of senescent cells¹⁴. We have added the above results in Figure 3D and 3E and have presented them in the results section, lines 138-147.

In addition to the data on γ H2AX expression, we have presented data on senescence-associated β -galactosidase (SA- β GAL) activity in FAPs in the CIM and AMI models, as well as in response to exercise, by staining with SPiDER- β Gal. SA- β Gal is a histological marker of senescent cells^{15,16}. Please see Figure 2B and 2C, and Figure S4.

5. In order to test the effects of exercise, the authors use forced downhill running, and paradigm designed to induce muscle damage and thus not relevant to any of the exercise paradigms that would be used in patients. If the goal is to test the effects of non-injurious physical activity (as would be the goal in humans), there are other protocols that would result in muscle exercise without damaging injuries.

Author Response: Thank you for your comment. Low-intensity exercise has been preferred in cases of chronic myopathy because of the fear that exercise will aggravate muscle inflammation, but a human study recently demonstrated that intensive exercise had positive effects in chronic myopathy ¹⁷. Transient inflammation after intensive exercise is one of the triggers of muscle regeneration ¹⁸, and therefore intensive exercise for chronic myopathy has attracted attention ¹⁷. Inflammation is required to change the intramuscular environment in chronic myopathy from a degenerative to a regenerative state ¹⁹, and non-injurious physical exercise can limit the effect of exercise to only a minor improvement in motor function ¹⁷. Hence, when investigating new myopathy treatments in humans, it will be beneficial to assess the effects of exercise associated with muscle damage rather than only non-injurious physical exercise. However, the detailed mechanisms of the regeneration or degeneration/fibrosis induced by intensive exercise remain unknown. Therefore, to clarify this issue we used downhill running to induce muscle damage, with the hope that this research will lead to new myopathy treatments that are both safe and effective.

Given these major concerns about the experimental design and data interpretation, it is difficult to draw any conclusions about effects of exercise (as occurs in humans) on muscle regeneration in response to acute or chronic injury, the role of FAPs as mediators of any exercise intervention, and whether AICAR is likely to be beneficial with or without exercise (except for injurious downhill running) in improving muscle regeneration

Other comments

1. In Fig. 1C, the authors stain for collagen and conclude that it is produced by FAPs. There is no data presented to support that conclusion.

Author Response: Thank you for your comment. We have deleted the conclusion regarding collagen production by FAPs because there were no supporting data. Please see the results section, lines 86-87.

2. In Fig. 2, the authors test for the sensitivity of FAPs in CMI and AMI to exogenous TNF-alpha. While indicative of susceptibility, these data do not demonstrate an actual increase in apoptosis in response to endogenous TNF-alpha in either model.

Author Response: Thank you for your comment. A previous study by Lemos et al. examined the effect of TNF-alpha on FAP apoptosis in vivo^{20,21}. TNF-alpha-expressing macrophages in acute muscle inflammation induced FAP apoptosis, whereas in chronic inflammation, TGF-beta-expressing macrophages resulted in FAP accumulation, suggesting that FAP apoptosis might be positively correlated with the concentration of TNF-alpha. To investigate this question, we tested FAPs in both AMI and CIM mice with varying doses of TNF-alpha in vitro. In the AIM model, the percentage of FAPs that underwent apoptosis was positively correlated with the TNF-alpha dose, but in the CIM model, the percentage of apoptotic FAPs did not vary with the dose. This suggests that in CIM mice, FAPs might maintain an anti-apoptotic phenotype regardless of the concentration of endogenous TNF-alpha. Regarding the response of FAPs to endogenous TNF-alpha, it was reported that FAP apoptosis occurred in AMI in the presence of TNF-alpha, but did not occur in vivo when TNF-alpha was inhibited by anti-TNF-alpha antibody; this phenomenon also occurred in vitro²⁰. These results were similar to our own in vitro findings. In addition to the results of this in vitro study, we have added data on immunostaining with active caspase-3 and PDGFR α in AMI and CIM mice (please see our response to your major concern 3).

3. The authors claim that FAPs in the CIM model acquire a “clearance deficiency”. This conclusion appears to be based on the persistence of FAPs in that model compared to the AMI model. However, is there any evidence to support that there is a clearance deficiency as opposed to a condition of persistent stimulation of activation and proliferation to account for the persistent number of FAPs?

In any chronic model, there is ongoing damage that might be expected to continue to stimulate FAP proliferation, resulting in sustained cell numbers without any clearance deficiency.

Author Response: Thank you for your comment. We agree that the statement about clearance deficiency was based on the persistence of FAPs in the CIM model compared to the AMI model. To provide additional evidence to support the conclusion that there is a clearance deficiency of CIM-FAPs, we have added new data on the mRNA expression of programmed death ligand 1 (PD-L1), PD-L2, and CD47 in FAPs. The clearance deficiency of diseased cells, e.g., tumor cells and vascular smooth muscle cells in atherosclerosis, is thought to be caused by impairment of immune surveillance machinery^{10,22,23}. We found that the mRNA expression of PD-L1/2 and CD47 was significantly up-regulated in CIM-FAPs compared to AMI-FAPs (Fig. 2I). These results support our hypothesis that there is a clearance deficiency of CIM-FAPs, because PD-L1/2 and CD47 produce anti-phagocytic signals, referred to as “don’t find me” and “don’t eat me” signals, respectively²³. Therefore, the CIM-derived FAPs expressing PD-L1/2 and CD47 readily accumulated as a result of escaping from immunoclearance, as well as due to proliferation resulting from ongoing damage in chronic inflammation. We have added this information in Figure 2I, and this point has been made in the results section, lines 115-124, and the discussion section, lines 339-353.

We also performed a phagocytosis assay using FAPs from Trp53(-/-) mice; of note, the latter showed low senescent factor expression, resistance to TNF- α stimulation-induced apoptosis, and activation of PD-L1 and CD47 mRNA expression. We found that H₂O₂-treated Trp53(-/-) FAPs showed a lower percentage of phagocytotic cells than H₂O₂-treated Trp53(+/+) FAPs. We have added this information in Figure 4H-4L, and this point has been made in the results section, page 12-13, lines 198-212.

References

1. Horn, A. *et al.* Mitochondrial redox signaling enables repair of injured skeletal muscle cells. *Sci Signal* **10**, eaaj1978 (2017).
2. Kozakowska, M., Pietraszek-Gremplewicz, K., Jozkowicz, A. & Dulak, J. The role of oxidative stress in skeletal muscle injury and regeneration: focus on antioxidant enzymes. *J. Muscle Res. Cell. Motil.* **36**, 377–393 (2015).

3. Childs, B. G., Baker, D. J., Kirkland, J. L., Campisi, J. & van Deursen, J. M. Senescence and apoptosis: dueling or complementary cell fates? *EMBO Rep.* **15**, 1139–1153 (2014).
4. Chen, Q. M., Liu, J. & Merrett, J. B. Apoptosis or senescence-like growth arrest: influence of cell-cycle position, p53, p21 and bax in H₂O₂ response of normal human fibroblasts. *Biochem. J.* **347**, 543–551 (2000).
5. Lassus, P., Ferlin, M., Piette, J. & Hibner, U. Anti-apoptotic activity of low levels of wild-type p53. *EMBO J.* **15**, 4566–4573 (1996).
6. Tavana, O. *et al.* Absence of p53-dependent apoptosis leads to UV radiation hypersensitivity, enhanced immunosuppression and cellular senescence. *Cell Cycle* **9**, 3328–3336 (2010).
7. Kataoka, M. *et al.* Down-regulation of bcl-2 is associated with p16INK4-mediated apoptosis in non-small cell lung cancer cells. *Oncogene* **19**, 1589–1595 (2000).
8. Hu, H. *et al.* P16 reactivation induces anoikis and exhibits antitumour potency by downregulating Akt/survivin signalling in hepatocellular carcinoma cells. *Gut* **60**, 710–721 (2011).
9. Chao, M. P., Weissman, I. L. & Majeti, R. The CD47-SIRP α pathway in cancer immune evasion and potential therapeutic implications. *Curr. Opin. Immunol.* **24**, 225–232 (2012).
10. Zou, W., Wolchok, J. D. & Chen, L. PD-L1 (B7-H1) and PD-1 pathway blockade for cancer therapy: Mechanisms, response biomarkers, and combinations. *Sci Transl Med* **8**, 328rv4–328rv4 (2016).
11. Mah, L.-J., El-Osta, A. & Karagiannis, T. C. gammaH2AX: a sensitive molecular marker of DNA damage and repair. *Leukemia* **24**, 679–686 (2010).
12. Sedelnikova, O. A., Rogakou, E. P., Panyutin, I. G. & Bonner, W. M. Quantitative detection of (125)IdU-induced DNA double-strand breaks with gamma-H2AX antibody. *Radiat. Res.* **158**, 486–492 (2002).
13. d'Adda di Fagagna, F. *et al.* A DNA damage checkpoint response in telomere-initiated senescence. *Nature* **426**, 194–198 (2003).
14. Wang, C. *et al.* DNA damage response and cellular senescence in tissues of aging mice. *Aging Cell* **8**, 311–323 (2009).
15. Hall, B. M. *et al.* Aging of mice is associated with p16(Ink4a)- and β -galactosidase-positive macrophage accumulation that can be induced in young mice by senescent cells. *Aging (Albany NY)* **8**, 1294–1315 (2016).
16. Dimri, G. P. *et al.* A biomarker that identifies senescent human cells in culture and in aging skin in vivo. *Proc. Natl. Acad. Sci. U.S.A.* **92**, 9363–9367 (1995).
17. Alexanderson, H. Physical exercise as a treatment for adult and juvenile myositis. *J. Intern. Med.* **280**, 75–96 (2016).

18. Sciorati, C., Rigamonti, E., Manfredi, A. A. & Rovere-Querini, P. Cell death, clearance and immunity in the skeletal muscle. *Cell Death Differ.* **23**, 927–937 (2016).
19. Eming, S. A., Wynn, T. A. & Martin, P. Inflammation and metabolism in tissue repair and regeneration. *Science* **356**, 1026–1030 (2017).
20. Lemos, D. R. *et al.* Nilotinib reduces muscle fibrosis in chronic muscle injury by promoting TNF-mediated apoptosis of fibro/adipogenic progenitors. *Nat. Med.* **21**, 786–794 (2015).
21. Tidball, J. G. & Wehling-Henricks, M. Shifts in macrophage cytokine production drive muscle fibrosis. *Nat. Med.* **21**, 665–666 (2015).
22. Kojima, Y. *et al.* CD47-blocking antibodies restore phagocytosis and prevent atherosclerosis. *Nature* **536**, 86–90 (2016).
23. Casey, S. C. *et al.* MYC regulates the antitumor immune response through CD47 and PD-L1. *Science* **352**, 227–231 (2016).